# In What Ways Are Deep Neural Networks Invariant and How Should We Measure This?

**Henry Kvinge**[1,2], **Tegan H. Emerson**[1,3,4], **Grayson Jorgenson**[1],
**Scott Vasquez**[1], **Timothy Doster**[1] , **Jesse D. Lew**[5]
[1]Pacific Northwest National Laboratory
[2]Department of Mathematics, University of Washington
[3]Department of Mathematics, Colorado State University
[4]Department of Mathematical Sciences at the University of Texas, El Paso
[5]National Geospatial-Intelligence Agency
`henry.kvinge@pnnl.gov`

## Abstract

It is often said that a deep learning model is "invariant" to some specific type of transformation. However, what is meant by this statement strongly depends on the context in which it is made. In this paper we explore the nature of invariance and equivariance of deep learning models with the goal of better understanding the ways in which they actually capture these concepts on a formal level. We introduce a family of invariance and equivariance metrics that allows us to quantify these properties in a way that disentangles them from other metrics such as loss or accuracy. We use our metrics to better understand the two most popular methods used to build invariance into networks: data augmentation and equivariant layers. We draw a range of conclusions about invariance and equivariance in deep learning models, ranging from whether initializing a model with pretrained weights has an effect on a trained model's invariance, to the extent to which invariance learned via training can generalize to out-of-distribution data.

## 1 Introduction

The notions of invariance and equivariance have been guiding concepts across a diverse range of scientific domains, from physics to psychology. In machine learning (ML) the concept of invariance is frequently invoked to describe models whose output does not change when its input is transformed in ways that are irrelevant to the task the model was designed for. For a dog or cat image classifier, for example, it is desirable for the model to be reflection invariant so that the same prediction is made whether or not the input image is reflected across its vertical axis. This type of invariance is useful in this model because although an image generally changes when reflected, whether it contains a dog or cat does not. Equivariance is used to describe models whose output changes in a manner that is aligned with the way that input is transformed. For example, as an image is rotated, the position of bounding boxes predicted by an object detector should also rotate (thus, the object detector should be rotation equivariant).

While mathematicians have developed rigorous theory that can describe invariance and equivariance, the amount of this that is actually used in the context of ML varies dramatically among different works. Within computer vision, research on relatively simple image transformations (i.e., rotations or translations) is often presented within a solid group-theoretic framework [6; 52; 9]. Other work dealing with more complex types of transformations such as changes in image background have (by necessity) needed to be more informal [51]. Furthermore, while model invariance and equivariance are frequently a central component in a broad range of ML research, limited effort has been put

into trying to measure them directly with the purpose of understanding the general invariance and equivariance properties of deep learning models. Rather, most works measure them indirectly through other ML metrics that align with the ultimate purpose of the model (e.g., loss or accuracy). While such a strategy makes sense when optimizing model performance is the primary objective, we miss an opportunity to better understand how and why deep neural networks work (or fail). The purpose of this paper is to propose a group-theoretic family of metrics, associated to an arbitrary group of symmetries $G$, which we call *$G$-empirical equivariance deviation* ($G$-EED), that quantify the $G$-equivariance (and $G$-invariance) of a model.

Informally, the $G$-EED of a model measures the extent to which it fails to be $G$-equivariant on a specific data distribution and with respect to a specific notion of distance in output space. This aligns with needs in ML where the user of a model may care only that their model is equivariant on data that the model will actually encounter in practice and not on any possible input. Further, since invariance is a special case of equivariance, $G$-EED also measures the extent to which a model fails to be invariant to the action of $G$. To show the breadth of the $G$-EED concept, we give a number of different ways it can be applied to measure different aspects of equivariance in a model. For example, we show how $G$-EED can be applied to a model's latent space representations to measure the extent to which a model extracts $G$-invariant features.

Finally, we use $G$-EED to answer a range of questions about invariance and equivariance in neural networks, with a focus on the two most popular ways of inducing invariance in these models: data augmentation and equivariant architectures. Some of the conclusions we draw from our experiments include the following. (1) Training with augmentation does not tend to induce invariance through learned equivariant layers; rather, invariance arises through some other mechanism. (2) Invariance learned through augmentation does generalize mildly to out-of-distribution data (e.g., common image corruptions), but apparent invariance seen for far out-of-distribution data (e.g., images from a completely different domain) could be the result of model insensitivity. (3) Invariance should not be assumed to correlate with model performance: models with random weights can be more invariant (in the usual mathematical sense) than models with learned or hard-coded invariance. (4) Models initialized with pretrained weights tend to have different invariance or equivariance properties than models initialized with random weights though whether these models are judged to be more invariant depends on the specific notion of distance one chooses to use. (5) Self-supervised models do not seem to be more invariant than supervised models except when they are trained with contrastive loss and augmentations from the relevant symmetry group $G$.

In summary, the contributions of this paper include:

- The introduction of $G$-empirical equivariance deviation, the first family of metrics that can rigorously and directly measure a range of notions of invariance and equivariance in deep learning models.

- A demonstration of the flexibility of $G$-EED, showing that it can be applied easily to a range of different components of a deep learning model to measure different forms of invariance and equivariance.

- Use of $G$-EED to answer a range of questions, shedding light on the extent to which neural networks are or are not invariant and equivariant.

## 2   Related Work

The literature on invariance and equivariance in machine learning can roughly be partitioned into two groups: those works that focus on how to build invariance and equivariance into a model or its components [27; 9; 52; 45; 10; 46; 19; 12; 37; 41; 2] and those works that focus on the theory behind invariance and equivariance [3; 6; 5; 34; 40].

Three common approaches are used to build invariance into deep learning models: data augmentation, feature averaging, and equivariant architectures. In this paper we focus on the first and third of these. Outside of a few ubiquitous layer types (such as standard translation equivariant convolutional layers), data augmentation is by far the most commonly used method, being a standard component of many training routines, particularly in computer vision. Since it has become a common procedure when training deep learning models, data augmentation research has expanded in many directions [12; 37; 41; 2].

The idea that $G$-invariance can be hardcoded into a model by combining multiple $G$-equivariant layers with data reduction layers (e.g., pooling) has a long history in deep learning. The most famous example of this idea is the conventional convolutional neural network (CNN) [32]. Since then, a multitude of other group equivariant layers have been designed including two-dimensional rotation equivariant layers [47; 50; 36; 38; 39], three-dimensional rotational equivariant layers [46; 11; 16], layers equivariant to the Euclidean group and its subgroups [45] (which we test against in this paper), and layers that are equivariant with respect to the symmetric group [33].

Our work is not the first to analyze various aspects of invariance in neural network models. Lyle et al. [34] analyzed invariance with respect to the benefits and limitations of data augmentation and feature averaging, presenting both theoretical and empirical arguments for using feature averaging over data augmentation. More recently, Chen et al. [6] presented a useful group-theoretic framework with which to understand data augmentation. Relevant to the present work, Chen et al. [5] introduced a notion of *approximate invariance*. Unlike that work, however, which focused on theoretical results related to data augmentation, this paper aims to introduce metrics that can be applied to modern deep learning architectures and answers questions about invariance from an empirical perspective. There are a number of existing works that proposed metrics aimed at measuring the extent to which a model is not equivariant (e.g. [8; 22; 18; 43; 49]). Our work differs from these in two ways: (1) we build general metrics based on basic group theory that are designed to work across different groups and datatypes and (2) unlike other works that use their metric to evaluate the equivariance of a specific model, we use our metrics to explore how models learn (or do not learn) to be equivariant generally.

Finally, a range of recent works have shown that even beyond the standard evaluation statistics (e.g., accuracy), invariance is an important concept to consider when studying deep learning models. For example, Kaur et al. [25] showed that lack of invariance can be used to identify out-of-distribution inputs. A further series of works investigated whether excessive invariance can reduce adversarial robustness [23; 24; 40]. All of this work reinforces one of the primary messages of this paper, that it is important to be able to measure invariance and equivariance directly in a model.

## 3 Quantifying Invariance and Equivariance

We begin this section by recalling the mathematical definitions of equivariance and invariance. We present these definitions in terms of the mathematical concept of a group, which formally captures the notion of symmetry [15].

Assume that $G$ is a group. We say that $G$ *acts* on sets $X$ and $Y$ if there are maps $\phi_X : G \times X \to X$ and $\phi_Y : G \times Y \to Y$ that respect the composition operation of $G$. That is, for $g_1, g_2 \in G$ and $x \in X$,

$$\phi_X(g_2, \phi_X(g_1, x)) = \phi_X(g_2 g_1, x),$$

with an analogous condition for $\phi_Y$. Whenever the meaning is clear, we simplify notation by writing $\phi_X(g, x) = gx$ (with an analogous convention for $\phi_Y$). A map $f : X \to Y$ is said to be *G-equivariant* if for all $x \in X$ and $g \in G$,

$$f(gx) = gf(x). \tag{1}$$

In the case where the map $\phi_Y$ is trivial so that $gy = y$ for all $g \in G$ and $y \in Y$, we say that $f$ is *G-invariant*. Thus, invariance is a special case of equivariance.

Assume that $f : X \to Y$ is a neural network where $X$ is the ambient space of input data and $Y$ is the target space. In many cases there is a natural way to factorize $f$ into a composition $f = f_2 \circ f_1$ where $f_1 : X \to Z$ is known as the feature extractor, $Z$ is the latent space of $f$, and $f_2 : Z \to Y$ is the classifier. For example, if $f$ is a ResNet50 CNN [20] then $f_1$ may consist of all residual blocks while $f_2$ would consist of the final affine classification and softmax layers. We say that machine learning model $f$ *extracts G-equivariant features* if $f_1$ is a $G$-equivariant function. This is an especially meaningful distinction in the context of transfer learning where invariance or equivariance can be transferred to a new task via the invariance or equivariance of $f_1$. Note that the definition of $G$-equivariant feature extraction requires a well-defined action of $G$ on $Z$, which may not be obvious in many cases. Because the trivial action is defined for any $G$ and $Z$, we can always ask whether $f$ *extracts G-invariant features*.

The following proposition provides some insight into how the invariance (or lack of invariance) of $f_1$ relates to the invariance (or lack of invariance) of $f$.

**Proposition 3.1.** Let $f : X \to Y$ be a function that decomposes into $f = f_2 \circ f_1$ where $f_1 : X \to Z$ and $f_2 : Z \to Y$. Suppose that $G$ acts on $X$, $Z$, and $Y$.

1. $f$ can be $G$-invariant even if $f_1$ and $f_2$ are not.

2. If $f_1$ is $G$-invariant, then $f$ is $G$-invariant.

A proof of Proposition 3.1 can be found in Section A.5. Note that Proposition 3.1.2 implies that if earlier layers of a network achieve invariance with respect to some group, then this invariance will persist into later layers. This may be seen as part of the justification of the common practice within the equivariant architectures community of building invariance through successive combinations of $G$-equivariant layers and pooling layers. Of course, the statement holds only when exact invariance is achieved. We see below that this is not generally the case.

## 3.1 Measuring equivariance

In this section we assume that both $X$ and $Y$ are vector spaces and the action of $G$ on both $X$ and $Y$ is linear. In all of our experiments we assume that the action of $G$ on $X$ and $Y$ is known. In the case that $f$ is also a linear map, the equivariance of $f$ can be checked directly by checking equivariance on a basis for $X$. By extension, the equivariance of many common types of neural network layers can be checked when these can be framed as linear maps (e.g., convolutional layers). However, there is no systematic procedure for checking the equivariance of a nonlinear function $f$. If $f$ is the composition of a sequence of functions $f_1, f_2, \ldots, f_n$ and we can check that each is equivariant, then we know that $f$ is equivariant, but we cannot prove that a function is not equivariant just by proving that each layer is not equivariant (this follows from Proposition 3.1.1). We are thus motivated to introduce a family of metrics that can be used to empirically measure the extent to which a function deviates from being equivariant on a data distribution $\mathcal{D}$ on $X$.

Since we assume in this section that the action of $G$ on $Y$ is linear, we can define the kernel of this action, $\ker(\phi_Y)$, which is a subgroup of $G$. $\ker(\phi_Y)$ consists of all those $g \in G$ such that $g$ acts as the identity on $Y$. For $x \in X$, we define $\hat{f}(x)$ to be the expected value of $f(gx)$ over $\ker(\phi_Y)$,

$$\hat{f}(x) := \mathbb{E}_{g \in \ker(\phi_Y)}(f(gx)) = \int_{g \in \ker(\phi_Y)} f(gx) d\mu \tag{2}$$

where $\mu$ is the usual normalized Haar measure on subgroup $\ker(G)$. Note that when $f$ is $G$-equivariant, then for each $g \in \ker(\phi_Y)$, $f(gx) = gf(x) = f(x)$ and hence $\hat{f} = f$.

Let $(\mathcal{D}, \nu)$ be a probability distribution on $X$ and let $m : Y \times Y \to \mathbb{R}_{\geq 0}$ be a distance function on $Y$. We can use $m$ to measure the extent to which $f$ deviates from being $\bar{G}$-equivariant by computing

$$\int_{\mathcal{D}} \int_{g \in G} m(f(gx), g\hat{f}(x)) d\mu dx \tag{3}$$

where this time $\mu$ is the normalized Haar measure on $G$. Note that the argument $m(f(gx), g\hat{f}(x))$ measures the extent to which (1) fails to hold across distribution $\mathcal{D}$, except that $gf(x)$ is replaced by $g\hat{f}(x)$. We use $\hat{f}(x)$ because it averages over all values in the orbit of $x$ (under $G$) that should map to $f(x)$. If $f$ were genuinely $G$-equivariant, all these values $f(gx)$ would yield $f(x)$. This choice is supported by the fact that it naturally interpolates between the two extreme cases: $G$ acts trivially on $Y$ (invariance) and $G$ acts faithfully on $Y$. In the former case $\hat{f}(x)$ is the average value of $f(gx)$ over all of $g \in G$ and in the latter case $\hat{f}(x) = f(x)$.

The proposition below proves that when the action of $G$ is faithful on $X, Y$, (3) being 0 is equivalent to $f$ satisfying (1) on a set of measure 1.

**Proposition 3.2.** Let $f : X \to Y$ be a continuous function, $G$ a group that acts linearly and faithfully on both $X$ and $Y$, and $m : Y \times Y \to \mathbb{R}_{\geq 0}$ a metric. Let $\mathcal{D}$ be a distribution on $X$. Then (3) is zero if and only if $f$ is $G$-equivariant almost surely, i.e., on a set of measure 1.

We provide a proof of Proposition 3.2 in Appendix A.6

To approximate (3) for real models and data where we always work with finite groups and finite samples of $\mathcal{D}$, we define the $G$-EED of $f$ with respect to $m$ to be

$$\mathcal{E}(f,G) := \frac{1}{|D||G|} \sum_{x \in D} \sum_{g \in G} m(f(gx), g\hat{f}(x)). \tag{4}$$

Note that since $G$ is a finite group (and hence discrete), the Haar measure turns into the usual counting measure.

In the remainder of this section we describe some specific types of $G$-EED that may be relevant to computer vision tasks. By convention, when using a distance function $m$ for which larger values of $m(x_1, x_2)$ indicates that $x_1$ and $x_2$ are "closer" (the opposite of a proper metric), we attach a negative sign to $m$. For example, rather than using cosine similarity, we use negative cosine similarity (e.g., (5)). This way, larger values of $G$-EED consistently indicate less invariance, regardless of the $m$ used.

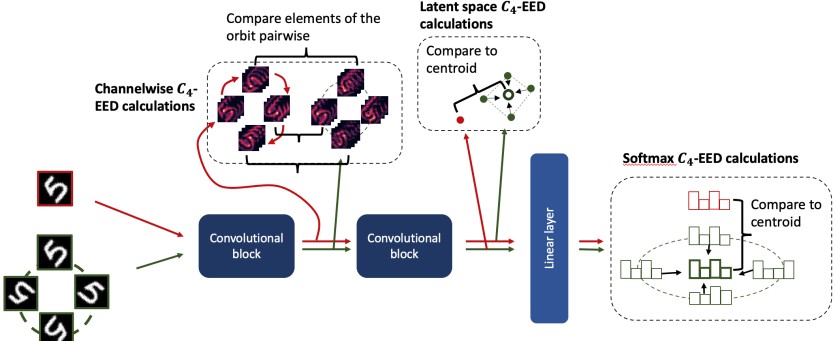

Figure 1: A diagram illustrating the different types of empirical equivariance deviation (EED) that we investigate in this paper for the rotation action of cyclic group $C_4$ on MNIST images.

**The channelwise $G$-equivariance of convolutional layer activations:** Throughout most layers of a CNN an individual image is represented as a 3-tensor. Let $f_\ell : X \to \mathbb{R}^{C_\ell \times H_\ell \times W_\ell}$ be the composition of the first $\ell$ layers of a CNN such that for input image $x \in X$, $f_\ell(x)$ is a $C_\ell \times H_\ell \times W_\ell$ tensor where the first dimension corresponds to the channels of the representation and the second and third dimensions correspond to the two spatial dimensions.

Suppose that $G$ is a finite group that acts on images and other 2-tensors (e.g., rotations, translations, and reflections). To understand the extent to which the first $\ell$ layers of the network are $G$-equivariant, we can measure the $G$-EED of $f_\ell$. Although there are numerous choices of $m$ for (3) that could be used to measure the channelwise difference between $f_\ell(gx)$ and $gf_\ell(x)$, we choose the following: let $S : \mathbb{R}^{H_\ell \times W_\ell} \times \mathbb{R}^{H_\ell \times W_\ell} \to [-1, 1]$ be the cosine similarity on individual channels treated as vectors in $\mathbb{R}^{H_\ell W_\ell}$. Write $[f_\ell(x)]_i$ for the $i$th channel of $f_\ell(x)$. Then we set

$$m(f_\ell(gx), g\hat{f}_\ell(x)) = -\frac{1}{C} \sum_{i=1}^{C} S([f_\ell(gx)]_i, g[\hat{f}_\ell(x)]_i).$$

That is, let $m$ be the negative of the average cosine similarity between individual channels of $f_\ell(gx)$ and $g\hat{f}_\ell(x)$. This gives:

$$\mathcal{E}_{\text{channel}}(f, G, \ell) := \frac{1}{|D||G|} \sum_{x \in D} \sum_{g \in G} m(f_\ell(gx), g\hat{f}_\ell(x)). \tag{5}$$

Note that since rotation, translation, and reflection groups all act faithfully on $\mathbb{R}^{H_\ell \times W_\ell}$, for these specific $G$, $\hat{f} = f$. We call this version of $G$-EED *channelwise $G$-EED*. This metric assumes that $G$ acts on each channel in a 3-tensor independently. It does not account for the more complicated setting where the action of $G$ either permutes or mixes channels of $\mathbb{R}^{C_\ell \times H_\ell \times W_\ell}$ in a non-trivial way. In this work we always assume that the action of $G$ on channels of $\mathbb{R}^{C_\ell \times H_\ell \times W_\ell}$ is identical to the action of $G$ on $X$ (up to differences in spatial scale and number of channels) except for the models

with equivariant architectures, where the action of $G$ is hardcoded to permute channels in a specific way. In these latter cases we match channels in accordances to the specifics of the architecture. We consider the case where the action of $G$ on $\mathbb{R}^{C_\ell \times H_\ell \times W_\ell}$ may permute channels in non-equivariant architectures in Appendix A.4.

**Invariance of latent space representations:** Consider a model $f$ consisting of feature extractor $f_1 : X \to \mathbb{R}^k$ and classifier $f_2 : \mathbb{R}^k \to \mathbb{R}^n$. If $G$ is some group that acts on $X$, we can ask whether $f_1$ extracts $G$-invariant features. Define $M$ to be either the average $\ell_2$-distance or cosine similarity between distinct points in the set $f_1(D)$. By setting $m$ to be the $\ell_2$-distance (respectively cosine similarity), we calculate the *latent space G-EED* as

$$\mathcal{E}_{\text{latent}}(f, G) := \frac{1}{M} \frac{1}{|D||G|} \sum_{x \in D} \sum_{g \in G} m(f_1(gx), \hat{f}_1(x)). \tag{6}$$

Note that because in this setting $G$ acts trivially on the latent space $\mathbb{R}^k$ (because of the invariance assumption), $\hat{f}_1(x) = \frac{1}{|G|} \sum_{g \in G} f_1(gx)$. We describe why the normalization term $M$ is necessary in Section A.7 of the Appendix. See also Figure 14. As we show in Section 4.1, it is often useful to consider latent space $G$-EED with respect to both Euclidean distance and negative cosine similarity.

**Invariance at the softmax layer of classifiers:**

While the two examples above focused on calculating invariance and equivariance at different intermediate layers of computer vision models, invariance of classification models is most commonly considered with respect to the final softmax output. In this paper we choose to use KL-divergence for $m$ because it is a distance function designed to handle probability distributions, which softmax outputs approximate. In particular,

$$\mathcal{E}_{\text{softmax}}(f, G) := \frac{1}{|D||G|} \sum_{x \in D} \sum_{g \in G} D_{KL}\big(f(gx), \hat{f}(x)\big).$$

We call this metric the *softmax G-EED*.

## 4 Understanding invariance and equivariance in deep learning

To prototype $G$-EED we experiment with the finite cyclic group $C_8$ that acts on images by $45°$ rotations. This symmetry is relevant to datasets where imagery does not have a preferred orientation. The datasets we use include *Rotated MNIST* which consists of randomly rotated MNIST digits [14] from classes 0-8 and *xView Maritime*, a chipped version of the xView object detector overhead dataset [31] composed of the nine maritime classes. We chose xView Maritime because it is a more real-world dataset that has an obvious symmetry (rotation) that is not hardcoded into standard CNNs. We describe these and other datasets we use in detail in Section A.2.

To reduce the computational burden, when calculating (4) we only used 50 randomly chosen examples from the corresponding dataset. We found that evaluating at more points did not substantially change the results. When not otherwise stated all invariance and equivariance measurements were calculated using points from the test set, not the training set. The normalization constant for latent space $G$-EED is calculated over 200 randomly chosen points.

### 4.1 Do networks trained with augmentation learn equivariant layers?

In this section we apply our metrics to standard CNNs and $C_8$-rotation equivariant, steerable CNNs ($C_8$-CNN) [45] to compare the invariance or equivariance of a model trained with augmentation with the invariance or equivariance of a model that is explicitly designed to learn invariance through progressive equivariant layers and pooling. We train 10 copies of each model type for 2,000 iterations on rotated MNIST. We apply random rotation augmentation to the models during training and all achieve an accuracy of over $98\%$ on the test set. By design, for any $1 \le k \le 6$, the first $k$-blocks of one of the $C_8$-CNN models will be approximately rotation equivariant (up to hardcoded permutation of channels which we take into consideration when applying our metrics). The standard wisdom is that, with pooling, this will lead to an invariant representation in the latent space.

Despite the fact that both families of models achieve similar performance, we note that their invariance and equivariance properties differ significantly. The channelwise $C_8$-EED of the composition of

the first $k$ blocks (where $1 \leq k \leq 6$) of the $C_8$-CNN is significantly higher than that of the CNN, indicating that (as claimed) the $C_8$-CNN is more equivariant with respect to individual channel rotations of $45°$ in both input and hidden representations (see Figure 7). Beyond that we see that equivariance decays for both models as input travels through additional blocks. This is particularly true for the $C_8$-CNN. We conjecture that this is likely caused by the accumulation of interpolation artifacts associated with rotation. Perhaps even more interesting is the fact that for no layer of either model do the channelwise $C_8$-EED values increase beyond a small jump at the beginning of training. This suggests that layerwise equivariance may not always be aligned with the learning objective and that learned invariance does not arise through the naive form of learned equivariance.

Both in terms of the Euclidean distance version of latent space $C_8$-EED and softmax $C_8$-EED, the $C_8$-CNNs show more invariance than the conventional CNNs (see Figure 2) but the difference is less dramatic than it is for channelwise $C_8$-EED. This suggests that the CNNs likely catch up in terms of $C_8$-invariance using mechanisms distinct from layerwise $C_8$-equivariance. Notable also is the fact that unlike channelwise $C_8$-EED, which did not indicate increased $C_8$-equivariance over the course of training, both latent space and softmax $C_8$-EED decreased with training (suggesting learned invariance at these layers).

Most surprisingly, in Figure 8 (left) we see that the $C_8$-CNNs are less invariant than the conventional CNNs with respect to the cosine similarity version of latent space $C_8$-EED. We conjecture two possible reasons for this. The first is based on the fact that $C_8$-CNNs are designed to be equivariant **on the nose**, **not up to scaling**. The structural constraint of exact equivariance may actually be misaligned with cosine similarity where vector magnitude no longer matters. From this perspective, projective representation theory, where symmetries are only taken up to scaling, might be a profitable direction of future research in equivariant architectures. The second possible explanation arises from Figure 8 (right) which shows that in terms of absolute cosine similarity (that is, without the normalization term $M$ in (6)), the $C_8$-CNN is more invariant. It may be that while $C_8$-CNNs indeed cluster orbits of points under the $C_8$ action closer together, the conventional CNNs are better at spreading other points further away in the latent space.

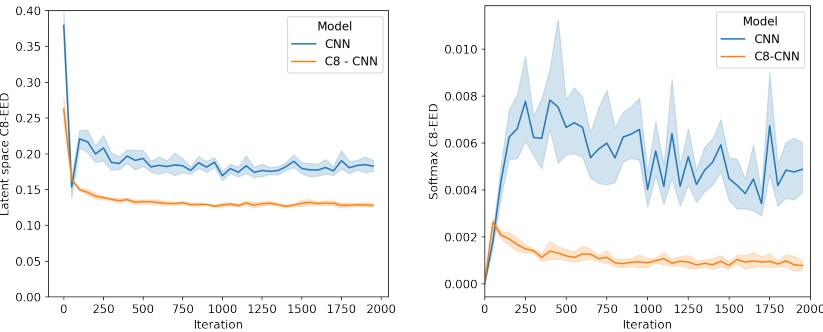

Figure 2: The latent space $C_8$-EED (left) and softmax $C_8$-EED (right) for conventional and $C_8$-equivariant CNNs ($C_8$-CNNs) [45] with respect to the rotated MNIST dataset. Both plots include $95\%$ confidence intervals. Lower values for both plots indicate more $C_8$-invariance. $95\%$ confidence intervals are over 10 randomly initialized models.

**Summary:** Equivariant architectures and nonequivariant architectures capture invariance in distinct ways. Nonequivariant networks do not learn layerwise equivariance (at least with respect to the naive $C_8$ action given by rotating individual channels in a hidden representation). Rather, their invariance seems to be captured by the final layers. The level of invariance measured in the latent space depends on the underlying metric $m$ which is used.

### 4.2 Does learned invariance hold for out-of-distribution data?

It was noted in Lyle et al. [34] that while augmentation may be satisfactory for many tasks, there is a risk that learned invariance will not extend to out-of-distribution (OOD) data. We investigate this question using $G$-EED metrics, studying the conventional CNN and $C_8$-equivariant CNN from Section 4.1 and a conventional CNN with untrained weights that can be used for comparison. Note

that because rotation invariance is at some level hardcoded into the architecture of the $C_8$-CNN, our default assumption is that the $C_8$-CNN models will tend to be more invariant on OOD data than models with learned augmentation. Images of example OOD data as well as dataset descriptions can be found in Appendix A.2.

The left plot in Figure 3 shows the latent space $C_8$-EED for the three model families that we tested. We can see that random weights extract significantly less invariant latent space representations compared to both the augmentation-trained CNNs and $C_8$-CNNs. Although this difference is not surprising for the $C_8$-CNNs, for the standard CNNs it indicates that augmentation does learn some rotation-invariant features that generalize to OOD data. Furthermore, while $C_8$-CNNs do exhibit more latent space invariance than the augmentation-trained CNNs, the difference is insignificant compared to the difference between augmentation-trained CNNs and CNNs with random weights. Note that in these OOD experiments, the normalization term $M$ in (6) is calculated using the OOD data (rather than the training or test set). This choice was made when considering situations in which a frozen network is used in combination with metric learning methods, as in Snell et al. [42], to solve a task with data drawn from a shifted distribution.

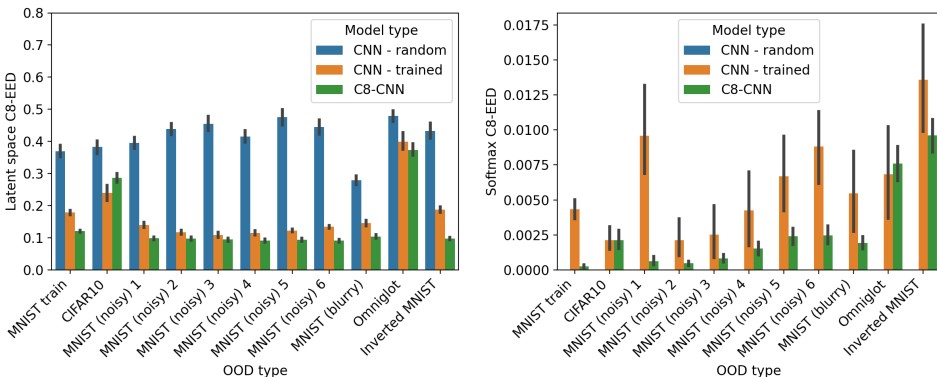

Figure 3: The latent space $C_8$-EED (left) and softmax $C_8$-EED (right) for conventional untrained CNNs, conventional CNNs trained on MNIST, and $C_8$-equivariant CNNs ($C_8$-CNNs) [45] trained on MNIST with respect to a range of in- and out-of-distribution datasets. Both plots include 95% confidence intervals over 10 randomly initialized models.

As a showcase of how a single model can display different types of invariance, the CNNs with random weights are far more invariant in terms of softmax $C_8$-EED than the trained $C_8$-CNN, which is more invariant than the trained CNN (right plot, Figure 3). We suspect that invariance of the CNNs with random weights arises from the fact that an untrained network tends to be insensitive to input. This observation is useful to keep in mind, especially as the concept of invariance is frequently treated as a net benefit to any model. Invariance is useful when it is aligned with other metrics more directly related to the task the model was designed for.

**Summary:** Our experiments suggest that augmentation trained networks do learn invariance that mildly generalizes to OOD data. A single model can display differing kinds of invariance (an untrained network does not extract invariant features but through insensitivity may make invariant predictions). In cases where invariance is critical and OOD data is expected, equivariant architectures may be a safer choice than training with augmentation.

## 4.3 How does the use of pretrained weights affect invariance and equivariance?

It has become standard practice in computer vision to initialize a model using weights trained on a large, diverse image dataset such as ImageNet [13]. It is reasonable to ask how such a strategy affects the invariance and equivariance of the resulting models. We choose four different image classifier architectures: ResNet50 [20], AlexNet [29], DenseNet121 [21], and LeViT192 [1]. For each of these architectures, we initialized five models with random weights and five models with weights from either the Torchvision package [35] (ResNet50, AlexNet, and DenseNet121) or Timm [48] (LeViT192). We then trained all of these models on the xView Maritime training set. Next,

we evaluated the latent space $C_8$-EED and softmax $C_8$-EED on the xView Maritime test set. It is important to note that pretrained weights, which were generated by training on ImageNet, might not be expected to have learned to extract fully rotationally invariant features. Indeed, the objects found in ImageNet mostly have a preferred orientation at which they generally are seen (there are very few instances of upside-down dogs or vehicles in ImageNet).

Figure 4 shows the latent space $C_8$-EED for all eight model types recorded every 200 iterations. We see that in this experiment and for this specific metric, models trained from scratch extract more invariant features in their latent space. Moreover, while the invariance of models trained from scratch tends to decrease slightly over the course of training (latent space $C_8$-EED increases), the invariance of models trained using pretrained weights tends to increase slightly. This may relate to ImageNet's lack of object classes that appear at a wide range of orientations. The models with pretrained weights may need to learn additional rotation invariance. Contrast that with the weights trained from scratch that may learn non-robust rotation invariant features early which are refined to less rotationally invariant features that better optimize the cross-entropy loss function later. This, along with the fact that models using pretrained weights tend to have slightly higher accuracy, points to a complicated dynamics between model robustness, model invariance, and model performance which likely requires further exploration. We also calculated the latent space $D_8$-EED (where $D_8$ is the order 16 dihedral group) for these same models and dataset. The results are recorded in Figure 10. We find the patterns to be fairly similar.

On the other hand, Figure 16 in the Appendix shows that, in terms of softmax $C_8$-EED, the invariance properties of pretrained networks vs. networks trained from scratch is somewhat more ambiguous.

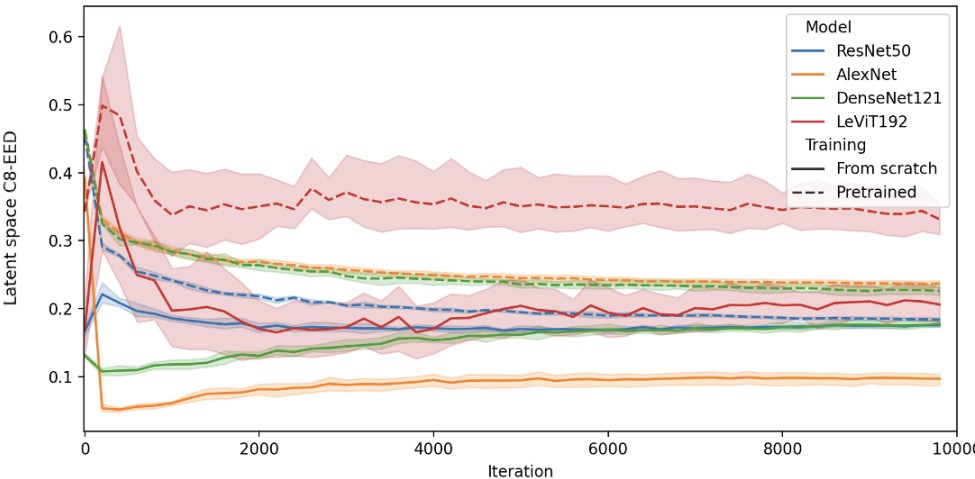

Figure 4: The latent space $C_8$-EED measured every 200 training iterations (for a total of 10,000 iterations) for a range of model architectures that were initialized with either random or ImageNet pretrained weights. Confidence intervals are over 5 randomly initialized models for 'from scratch' and 5 independent trainings for 'pretrained'.

**Summary:** Our experiments make it clear that the way in which invariances are learned differs between networks using pretrained weights and networks trained from scratch.

### 4.4   The invariance of supervised and self-supervised models

In an additional experiment, we compared the invariance of supervised models to the invariance of self-supervised models. To do this we evaluated four different convolutional neural networks in one of the simplest possible settings where $G$ is the order 2 group of reflections across the vertical axis. The models we chose to test were a standard ResNet50 (we used the Torchvision pretrained weights) [20; 35], a Big Transfer ResNet50V2_101 [26] (with weights from 'PyTorch Image Models' [48]), a ResNet50 trained using the self-supervised method DINO (we used the author's pretrained weights) [4], and a ResNet50 trained using the self-supervised method SimCLR [7] (we used weights from [44]). Note that the first two models utilized supervised training (with different size training sets)

and the second two utilized self-supervised training techniques. Further, SimCLR is trained with a contrastive loss that directly optimizes for invariance of input to specific transformations (including reflection). On the other hand, DINO uses 'knowledge distillation', without a contrastive loss. We choose to concentrate on latent space $G$-EED in this experiment.

Our results can be found in Figure 5. Note that since all of these models have a 2048-dimensional latent space, we can directly compare them. This is something we could not do in Section 4.3. We see that in terms of both versions of latent space $G$-EED (that which uses Euclidean distance and that which uses cosine similarity), the model trained using SimCLR has significantly more invariance. This is not surprising given that the contrastive loss that SimCLR uses explicitly optimizes for similarity between an image and its reflection. DINO, which is also trained in a self-supervised manner, but which does not explicitly use the contrastive loss is less invariant (how it compares to the two supervised models depends on which metric is used).

**Summary:** Use of contrastive loss can improve invariance but otherwise there is not strong evidence that self-supervision increases model invariance.

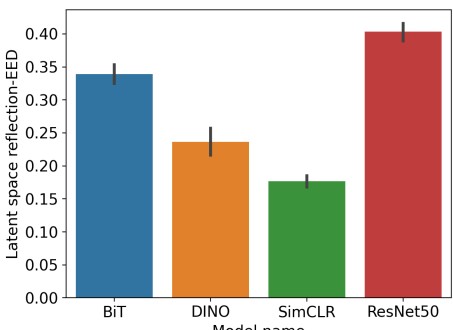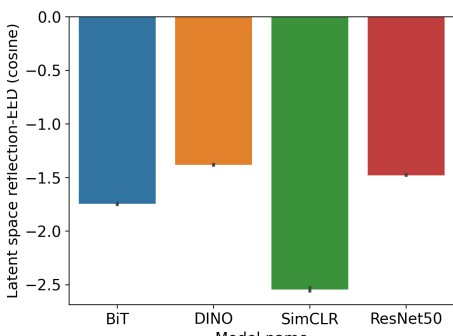

Figure 5: Latent space $G$-EED for two models trained using supervised methods (BiT and ResNet50) and two models trained using self-supervised methods (DINO and SimCLR), where $G$ is the order 2 group of reflections across the vertical axis. SimCLR is the only model trained with contrastive loss. All models use some version of the ResNet architecture. **(Left)** The standard latent space $G$-EED metric. **(Right)** $G$-EED where cosine similarity is used instead of Euclidean distance. Error bars represent $95\%$ confidence intervals over 100 random inputs from the ImageNet test set.

# 5   Limitations and Conclusion

Although the $G$-EED family of metrics that we introduce is flexible in many ways, it is not capable of capturing notions of invariance that do not have a known underlying symmetry group (e.g., changes in image background), do not come from the action of a group (e.g., scaling corresponding to a semigroup action), or (for the purpose of measuring equivariance of internal representations) do not have a known action on the hidden activations of a model (e.g., changes in color). An important next step would be to bring our metrics to these broader notions of invariance and equivariance. It would be particularly valuable if our metrics could be modified so that they can measure equivariance without explicitly specifying a group action in a hidden layer. This would enable us to measure $G$-EED for emergent types of learned equivariance where the action of $G$ on the output space is not specified by the user.

In this paper we described a novel family of metrics meant to measure equivariance in deep learning models. We use these metrics to suggest answers to questions related to the extent to which neural networks are or are not invariant or equivariant to different types of transformations. Surprisingly, while invariance and equivariance are fundamental topics within machine learning, this is one of the first works (to our knowledge) that tries to broadly measure these properties empirically in modern neural networks. We hope that this work will open a conversation on this important topic that will ultimately lead to a better understanding of why deep learning models behave the way that they do.

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
