# A  Appendix

## A.1  Experimental Details

All our models were trained on an Nvidia A100 using the Adam optimizer. Batch size, learning rate (LR), and weight decay hyperparameters are summarized in the table below. Recall from Sections 4.1 through 4.3 that we train the CNNs and $C_8$-CNNs (architecture description below) on rotated MNIST and train AlexNet, ResNet50, DenseNet121, and LeViT192 on Maritime xView. Hyperparameters were chosen based on an informal search. Running our experiments with a more comprehensive hyperparameter search would be a worthwhile future exercise since it might yield insight into how different hyperparameters affect model invariance and equivariance.

The standard CNN and $C_8$-CNN used in Section 4.1 consist of 6 convolutional blocks, each containing a standard convolutional layer, a batch norm, a ReLU nonlinearity, and a pooling layer. The $C_8$-CNN models have an analogous 6 block structure with the difference that each block uses $C_8$-rotation equivariant, steerable convolutional layers and a so-called inner batch norm (a batch norm adapted for equivariant frameworks [45]). Both model types have a final linear layer.

Table 1: Experimental hyperparameters.

|            | Batch size | LR | Weight Decay |
|---|---|---|---|
| CNN | 64 | $5 \times 10^{-4}$ | $1 \times 10^{-5}$ |
| $C_8$-CNN | 64 | $5 \times 10^{-4}$ | $1 \times 10^{-5}$ |
| AlexNet | 64 | $5 \times 10^{-5}$ | $1 \times 10^{-5}$ |
| ResNet50 | 64 | $5 \times 10^{-4}$ | $1 \times 10^{-5}$ |
| DenseNet121 | 64 | $1 \times 10^{-5}$ | $1 \times 10^{-5}$ |
| LeViT192 | 64 | $5 \times 10^{-5}$ | $1 \times 10^{-5}$ |

## A.2  Dataset Details

The datasets used in our experiments include (i) *rotated MNIST*, which consists of images from MNIST rotated by random angles [14] (MNIST is covered by a Creative Commons Attribution-Share Alike 3.0 license) and (ii) *xView maritime*, a classification dataset we constructed by cropping the bounding boxes of all maritime vessel classes found in xView [31], an overhead imaging dataset (xView is covered by an Attribution-Noncommercial-ShareAlike 4.0 International (CC BY-NC-SA 4.0) license). In our version of rotated MNIST we excluded the class 9 so that we could remove the drop in model accuracy resulting from the similarity of a 9 and an upside-down 6. We apply a circular mask to all images before using them as input to a model so that invariance is not broken by the artifacts introduced by rotating a rectangular image by $\theta$ degrees where $\theta \neq 0°, 90°, 180°, 270°$. See the bottom right image in Figure 6 for an example of this.

Below we describe our OOD datasets for Section 4.2.

- **MNIST Train:** The original MNIST training images (with class 9 excluded).
- **CIFAR10:** Grayscale versions of the CIFAR10 images [28] (CIFAR10 is covered under an MIT license).
- **MNIST (Noisy) 1-6:** MNIST images perturbed by noise randomly sampled uniformly from intervals (1) $[0, 0.2]$, (2) $[0, 0.4]$, (3) $[0, 0.6]$, (4) $[0, 0.8]$, (5) $[0, 1.0]$, and (6) $[0, 1.2]$. Notice that any of these noise perturbations can push pixels outside of image bounds. We generally assume that images with more noise are farther OOD.
- **MNIST (blurry):** MNIST images with Gaussian blur applied with a size $7 \times 7$ kernel with standard deviation .5.
- **Omniglot:** Another image classification dataset that contains characters [30] (Omniglot is covered under an MIT license).
- **Inverted MNIST:** MNIST where each pixel has had the transformation $f(x) = 1 - x$ applied to it. This inverts the intensity of pixels to make the MNIST dataset look more like Omniglot (the background has high intensity and the digits have low intensity).

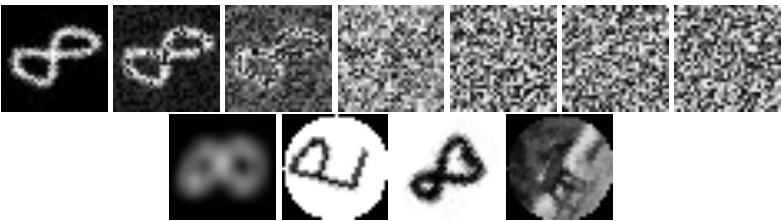

Figure 6: Examples of the OOD datasets that we used to evaluate our models. From left to right and top to bottom: MNIST train, MNIST (noisy) 1–6, MNIST (blurry), Omniglot, inverted MNIST, and CIFAR10 (grayscale).

### A.3    How does invariance in a model's latent space change when augmentation is or is not used?

As in Section 4.3, we focus on the case where $G = C_8$ (the group generated by $45°$ rotations) and the dataset xView Maritime. We trained 5 ResNet50s, 5 AlexNets, and 5 DenseNet121s with and without rotation augmentation respectively. We then measured latent space $C_8$-EED in all models. Our results can be found in Figure 11. We find that models trained without augmentation indeed had consistently lower $C_8$-EED both when we used our standard Euclidean distance metric and when we used the cosine similarity version. These experiments serve as evidence that our metrics are measuring the properties that we think they are measuring.

### A.4    Does learned equivariance involve re-ordering of tensor channels?

Suppose that the function $f_\ell : X \to \mathbb{R}^{C_\ell \times H_\ell \times W_\ell}$ corresponds to the first $\ell$ layers of a network, terminating with a 3-tensor which has $C_\ell$ channels of height $H_\ell$ and width $W_\ell$. Suppose that $G$ is a group that acts both on the input space $X$ and the hidden space $\mathbb{R}^{C_\ell \times H_\ell \times W_\ell}$. The channelwise $G$-EED metric that we proposed in this paper, assumes that when we compare channels from $f_\ell(gx)$ with channels from $gf_\ell(x)$, we should assume the trivial bijection. That is, we should compare the first channel of $f_\ell(gx)$, $[f_\ell(gx)]_1$, with the first channel of $gf_\ell(x)$, $[gf_\ell(x)]_1$, the second channel $[f_\ell(gx)]_2$ with the second channel $[gf_\ell(x)]_2$, etc.

On the other hand, in many equivariant CNN's, the group action of $g$ on $f_\ell(x)$ not only changes individual channels, it also permutes their order. It is reasonable to ask whether CNNs trained with augmentation might learn some similar "emergent" structure not only within individual channels, but also among them. The channelwise $G$-EED metric would likely not detect this kind of equivariance. In this section we examine this possibility. Though we do not disprove its existence, we run several preliminary experiments that suggest that this is not likely. This is an area that would benefit from additional study.

In our preliminary investigation, we focus on the filters at each layer (rather than input and output). We chose to do this based off of the observation that in equivariant CNNs that utilize the regular representation and which are designed to display the phenomenon we are looking for, filters come in entire orbits. That is, if $w$ is a filter in a layer, then so is each element in the orbit $Gw$ (i.e., the orbit of $w$ under the action of $G$). In Figure 12 for example, four filters of a $C_4$-equivariant convolutional layer [45] are displayed. The filters represent an orbit under the action of all 90-degree rotations (e.g., $C_4$).

As a first step towards identifying structured equivariance in augmentation trained CNNs, we compared all channels of each pair of filters at a given convolutional layer of an AlexNet CNN. However, comparing all the possible pairings is combinatorially prohibitive, even for simple architectures like AlexNet. In Figure 13, we plot the metric

$$\min_{\substack{j \neq i;k \\ g \in C_4}} ||g(w_{\ell,i,t}) - w_{\ell,j,k}||_{\ell_2}, \tag{7}$$

where $w_{\ell,i,t}$ is the height $H'_\ell$ and weight $W'_\ell$ 2-tensor obtained by taking channel $t$ from filter $i$ in layer $\ell$ and $g(w_{\ell,i,t})$ is the action of group element $g$ on $w_{\ell,i,t}$. It this case simply rotating by a multiple of $90°$. We report the average of this metric over all convolutional layers of four different

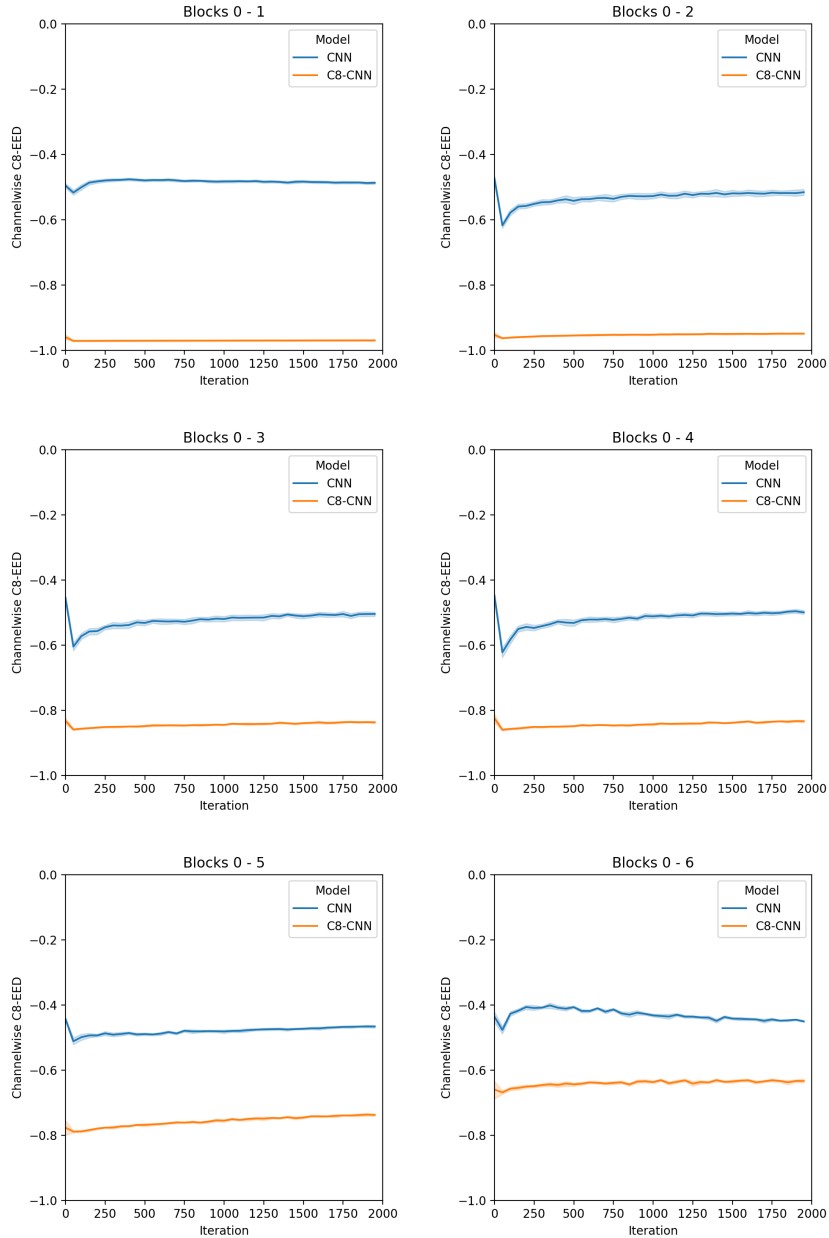

Figure 7: The channelwise $C_8$-EED for the composition of various layers in 10 conventional CNNs and 10 $C_8$-equivariant CNNs ($C_8$-CNN) [45] trained on the rotated MNIST dataset. Both plots include $95\%$ confidence intervals. Smaller values indicate more $C_8$-equivariance.

AlexNet models trained on MNIST: AlexNet trained from scratch without rotation augmentation, AlexNet trained with rotation augmentation but no pretraining, AlexNet with both pretraining and rotation augmentation, and finally Alex with $C_8$-equivariant layers [45]. Random $i$ and $t$ are selected for each computation, and for computational simplicity we only evaluated on the subgroup $C_4$ of $C_8$ generated by 90-degree rotations.

A network where some filters are rotations of others would be expected to achieve a value of $0$ for (7). Indeed, we can see that this is what happens to the $C_8$-equivariant network. We see that the other networks do not achieve zero. Indeed, with the exception of the pretrained AlexNet, for which (7)

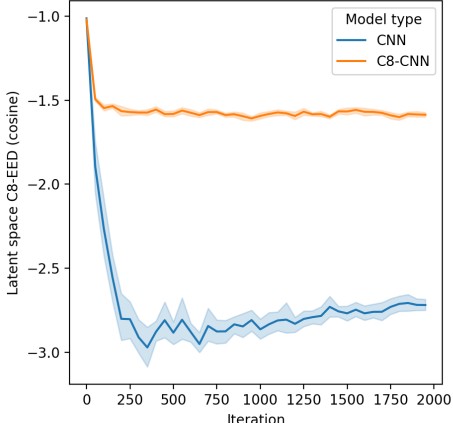
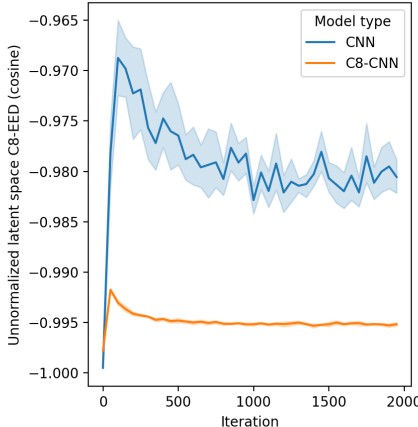

Figure 8: **(Left)** The cosine similarity version of latent space $C_8$-EED for 10 conventional CNNs and 10 $C_8$-CNNs all trained on rotated MNIST. Surprisingly, the $C_8$-CNNs are **less** invariant with respect to this metric. We speculate as to why this might be in Section 4.1. **(Right)** One of our hypotheses is driven by plotting the unnormaized version of this metric which suggests that $C_8$-CNNs clusters points in a single $C_8$ orbit closer than the CNNs, but do not scatter other points as far as the CNNs do.

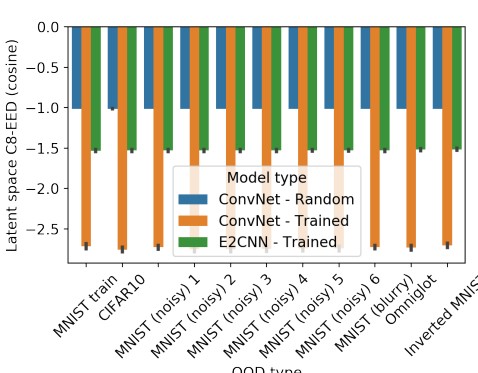
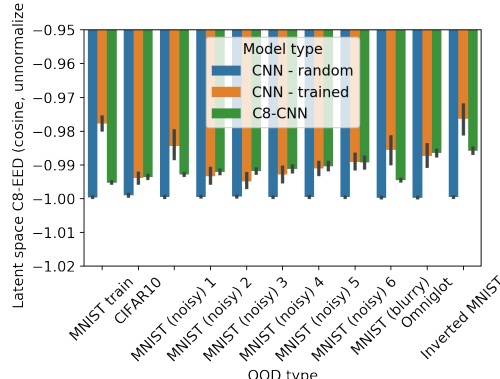

Figure 9: The cosine similarity version of latent space $C_8$-EED (**normalized on the left**, **unnormalized on the right**) for conventional untrained CNNs, conventional CNNs trained on MNIST, and $C_8$-equivariant CNNs ($C_8$-CNN) [45] trained on MNIST with respect to a range of in- and out-of-distribution datasets. Both plots include 95% confidence intervals. **Lower values indications more $C_8$-invariance.** As can be seen, without normalization, the trained conventional CNNs and the $C_8$-CNNs are often comparable. On the other hand, after normalization the conventional CNNs show much lower latent space $C_8$-EEG, indicating more $C_8$-invariance. The differences between these two plots seems to indicate that the conventional CNNs have learned to better separate orbits of points rather than learning to cluster orbits together more tightly.

increases slightly over training, the other non-equivariant networks do not change significantly at all. This indicates that this form of emergent equivariance does not emerge in this example.

## A.5    A Proof of Proposition 3.1

*Proof.*        1. To prove this, consider the function $f : \mathbb{R}^2 \to \mathbb{R}$ defined such that for $(x, y) \in \mathbb{R}^2$

$$f(x, y) = \frac{x + y}{2}.$$

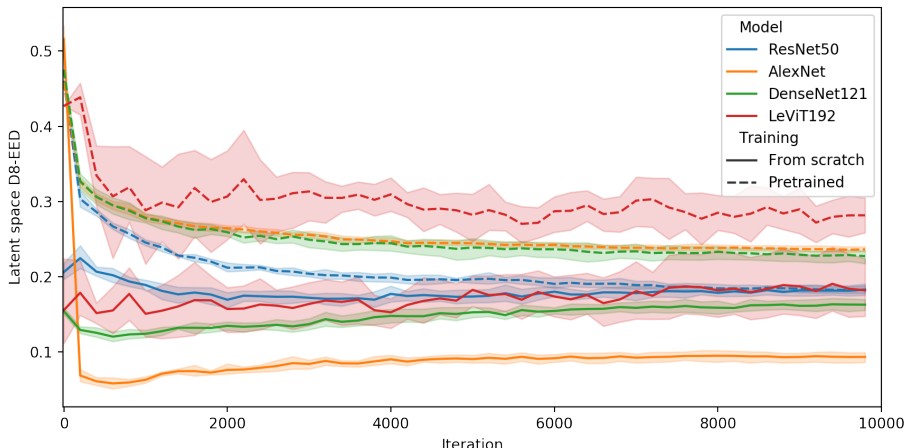

Figure 10: The latent space $D_8$-EED measured every 200 training iterations (for a total of 10,000 iterations) for a range of model architectures that were initialized with either random or pretrained weights generated using ImageNet. Note that $D_8$ is the dihedral group of order 16 generated by a $45°$ angle and a reflection across an axis.

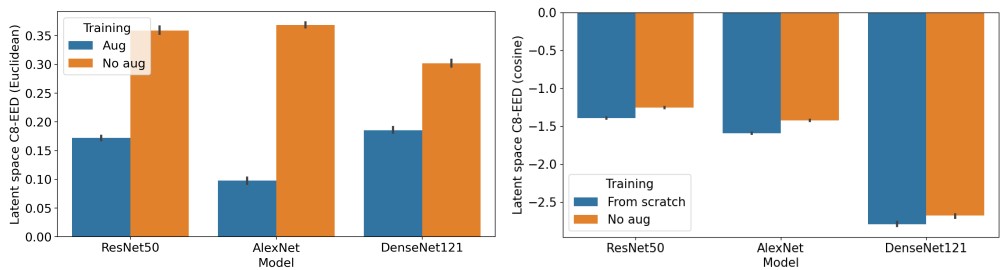

Figure 11: Latent space $C_8$-EED on xView Maritime for models trained with and without rotation augmentation. **(Left)** The standard latent space $C_8$-EED metric. **(Right)** The alternative version of latent space $C_8$-EED where cosine similarity is used instead of Euclidean distance. Note that different models are not necessarily comparable since their latent spaces can have different dimension. (**smaller means more invariant to rotation.**)

The order 2 cyclic group $\mathbb{Z}_2 = \{1, \sigma\}$ acts on $\mathbb{R}^2$ by permuting coordinates. That is, the only nonidentity element $\sigma$ acts on $\mathbb{R}^2$ by sending $(x, y) \mapsto (y, x)$. It is clear that $f$ is $\mathbb{Z}_2$-invariant.

However, note that if $f_1 : \mathbb{R}^2 \to \mathbb{R}^2$ is defined by $f_1(x, y) = (2x, y)$ and $f_2 : \mathbb{R}^2 \to \mathbb{R}$ is defined by

$$f_2(x, y) = \frac{x + 2y}{4},$$

then $f = f_2 \circ f_1$. But neither $f_1$ nor $f_2$ is $\mathbb{Z}_2$-equivariant to the actions described above.

2. This proof follows easily from the definitions. Suppose that $f_1$ is $G$-invariant, then for any $x \in X$ and $g \in G$, $f_1(gx) = f_1(x)$. Hence

$$f(gx) = f_2(f_1(gx)) = f_2(f_1(x)) = f(x).$$

$\square$

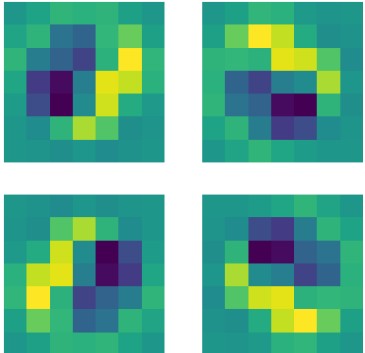

Figure 12: Four $7 \times 7$ filters from a randomly initialized $C_4$-equivariant convolutional layer. Note that this is the orbit of one of these filters under the rotation action of $C_4$.

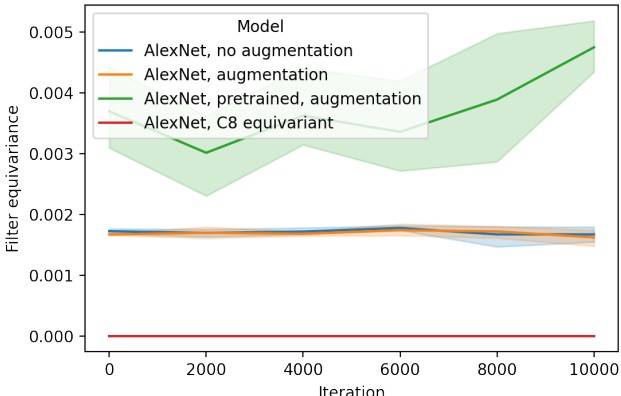

Figure 13: The average minimum $\ell_2$ norm differences (7), between pairs of individual filters (one with group element $g \in C_4$ applied and one not) from the convolutional layer of AlexNets. Augmented training does not appear to cause emergent equivariance of the kind seen in hardcoded equivariant architecture.

### A.6   A Proof of Proposition 3.2

*Proof.* First note that for fixed $x$ and $g$, $m(f(gx), g\hat{f}(x)) = 0$ if and only if $f(gx) = gf(x)$. This follows from the fact that since $G$ acts faithfully on $Y$, $\hat{f}(x) = f(x)$ and from the fact that $m$ is a metric. First assume that $m(f(gx), g\hat{f}(x)) = 0$. Then, since $m$ is a metric, $f(gx) = g\hat{f}(x) = gf(x)$ giving the desired result. Next, if $f(gx) = gf(x) = g\hat{f}(x)$, then it again follows that $m(f(gx), g\hat{f}(x)) = 0$.

Next we recall the basic fact [17, proposition 2.16] from measure theory, which states that if $h : W \to [0, \infty]$ is a non-negative measurable function on measure space $(W, \nu)$, then

$$\int_W h = 0 \Leftrightarrow h = 0 \ \text{ almost everywhere.} \tag{8}$$

Since $\hat{f}(x) = f(x)$, let $W = X \times G$ with $\xi = \nu \times \mu$ be the product measure on $W$ generated by Haar measure $\mu$ on $G$, and the probability measure $\nu$ associated with $\mathcal{D}$. Note that $f$ is measurable by virtue of being continuous, $G$ acts linearly and hence is measurable, and $m$ is measurable by virtue of being a metric. It follows that the function $h : X \times G \to [0, \infty]$ defined by

$$h(x, g) = m(f(gx), gf(x))$$

is measurable. The result then follows from the observation that $m$ is non-negative (and hence $h$ is) and from (8). That is, if (3) is zero, then since $m(f(gx), g\hat{f}(x))$ is non-negative, then (8) tells us that $m(f(gx), g\hat{f}(x)))$ is zero almost-everywhere with respect to measure $\xi$. On the other hand, (8) also says that if $m(f(gx), g\hat{f}(x))$ is zero almost everywhere (including the case where this term is zero everywhere), then (3) is equal to zero. $\square$

## A.7 The Reason for Normalization of Latent Space $G$-EED

We normalize $\mathcal{E}_{\text{latent}}(f, G)$ by $M$ because when we use $\ell_2$-distance to compute $G$-EED without normalization, it is sensitive to scaling in a way that is unrelated to downstream task performance. To illustrate, note that if $f = f_2 \circ f_1$ is a model with feature extractor $f_1$ and classifier $f_2$, then by scaling $f_1$ by $c > 1$, the unnormalized latent space $G$-EED increases, indicating a decrease in $G$-invariance. More precisely, if $f_1' = cf_1$, then $\mathcal{E}(f_1', G) = c\mathcal{E}(f_1, G)$. However, if we set $f_2' = \frac{1}{c}f_2$, then $f_2' \circ f_1' = f = f_2 \circ f_1$. Thus, $f_1$ can be made to have arbitrarily large unnormalized latent space $G$-EED while the model $f$ itself remains constant. A different but equally illustrative example is visualized in Figure 14. Two feature extractors have identical unnormalized latent space $C_8$-EED for a single rotation orbit of an image of a 4 (average distance between blue points and their centroid), but the second feature extractor closely clusters points in the orbit relative to other instances from MNIST, while the first feature extractor does not. We would argue that in most cases the second feature extractor should be called *more invariant* with respect to this particular task. Normalization mitigates this issue, giving a more reasonable notion of latent space invariance in the context of machine learning.

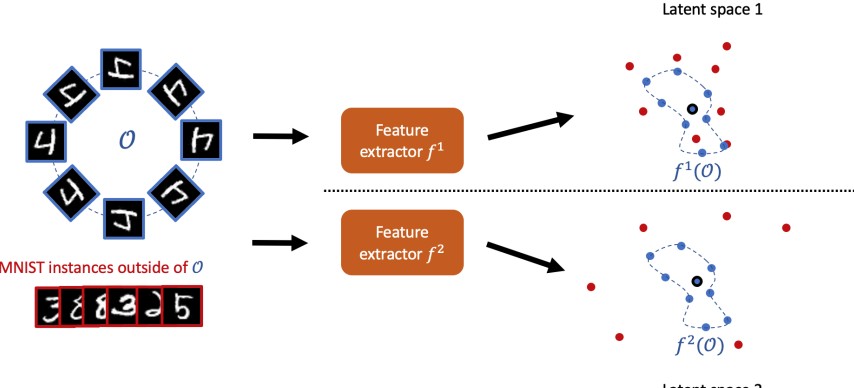

Figure 14: This illustration shows why it is important to normalize the latent space $G$-EED by the average distance between random pairs from the dataset. The set $\mathcal{O}$ (blue) is the orbit of an image of a '4' under the rotation action of $C_8$. Two feature extractors $f^1$ and $f^2$ map $\mathcal{O}$ as well as other unrelated images (red) from MNIST [14] to a latent space. While points from $f^1(\mathcal{O})$ and $f^2(\mathcal{O})$ both have the same average distance to their centroid, $f^2(\mathcal{O})$ clusters closely together relative to other points from the dataset. $f^1(\mathcal{O})$ is mixed with instances not belonging to $f^1(\mathcal{O})$. We would argue that $f^2$ extracts more invariant features.

## A.8 Channelwise $C_8$-EED for out-of-distribution datasets

In this section we provide Figure 15, which shows the channelwise $C_8$-EED for collections of: CNN models with random weights, CNN models that have been trained on MNIST, and $C_8$-CNN models that have been trained on MNIST.

In Figure 15 we show the channelwise $C_8$-EED at layers 2 and 5 for each of the model types. Unsurprisingly, the $C_8$-CNN has high $C_8$-equivariance for all layers, whereas both the trained and untrained CNNs have substantially lower $C_8$-equivariance. Across layers, the channelwise $C_8$-EED remains fairly constant across datasets for the untrained models. However, the channelwise $C_8$-EED for early layers of the trained CNNs differs across datasets (with equivariance on OOD datasets

generally being less than the equivariance on the training set MNIST). In later layers, however, the difference in $C_8$-equivariance between MNIST and an OOD dataset is negligible (Figure 15, right). This suggests that CNNs do learn some minimal amount of equivariance at early layers, but this equivariance either dissipates or becomes undetectable at later layers. Equivariance in earlier layers is also tied to image content. Images with high frequency signals tend to differ more significantly when rotated (with at the extreme end, a constant valued image unchanged with rotation).

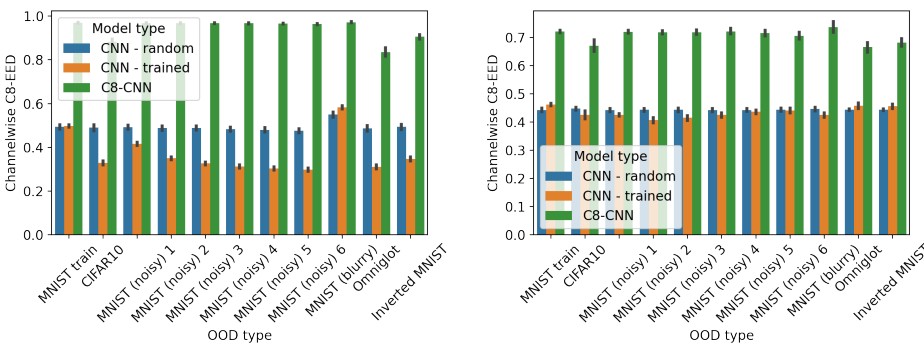

Figure 15: The channelwise $C_8$-EED for the first two (left) and first five (right) convolutional blocks of a CNN with random weights, a CNN with weights trained on MNIST, and a $C_8$-CNN with weights trained on MNIST. Error bars indicated $95\%$ confidence intervals.

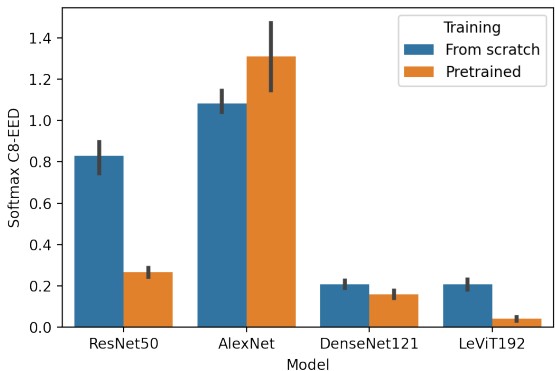

Figure 16: The softmax $C_8$-EED after $10,000$ iterations of training on xView maritime for a range of model architectures initialized with either random or pretrained weights generated using ImageNet. The $95\%$ confidence intervals are represented by the black bars.