# OpenReview forum: "In What Ways Are Deep Neural Networks Invariant and How Should We Measure This?"
_NeurIPS.cc/2022/Conference — NeurIPS 2022 Accept_

### Official Review · Reviewer_9Wpf · 2022-07-05

**Rating:** 7
**Confidence:** 4
**Soundness:** 3 good
**Presentation:** 4 excellent
**Contribution:** 3 good

**Summary:**

The paper introduces a metric for measuring invariance/equivariance of a model wrt a certain symmetry group and data distribution, called G-empirical equivariance deviation. This metric is used to compare to what extent equivariant models and models trained with data augmentation are invariant/have equivariant feature maps, and also to assess to what extent invariance/equivariance holds for OOD data.The paper also compares the extent of invariance/equivariance for using pretrained weights & fine tuning vs training from randomly initialized weights.

**Questions:**

Questions & Suggestions:
- It's confusing that the channelwise G-EED is different from softmax & latent space EED, in that a higher value means the model is more equivariant. Perhaps it's more consistent to use negative cosine similarity to be consistent with other metrics.
- Why is L2 distance used for latent space EED whereas cosine similarity is used for channelwise EED? If cosine similarity is also used for the former, it might not be necessary to normalizie by M, the average L2 distance between points.
- Figure 2 is used as evidence to claim "learned invariance does not arise through learned equivariance". It would be more convincing to show the plot for each value of k between 1 and 6 as opposed to just 2 & 5. As you mentioned in Prop 3.1, it could be that the first 5 layers aren't equivariant but all 6 conv layers are equivariant.
- In line 277-278, you mention that "invariance is useful when it is aligned with other metrics more directly related to the task the model was designed for". This makes me curious as to what would happen if you use each EED metric as a regulariser for a CNN - would it help of harm generalization?
- (minor) Figure 4 - x axis shifted by one.


**Strengths And Weaknesses:**

Strengths
- the paper is very clearly written, and does a good job of explaining concepts and motivation.
- it formalizes much needed metrics for measuring equivariance and invariance.
- the summaries at the end of each experimental section was useful to recap the conclusions of each experiment.
- Although the empirical results of this paper are not necessarily surprising and quite aligned with my intuition, it is useful to have such intuitions validated by empirical evidence (e.g. nonequivariant networks not learning layerwise equivariance, models trained from scratch are better at extracting invariant features)

Weaknesses
- the equivariance error metric in the paper is very much related to those used in various previous works on group equivariance, so not entirely novel. Search for 'equivariance error' in e.g. [1-3]. It would be good to cite these papers and mention that similar metrics have been used in the past, but all with slight differences. I still think there is value in formalizing such a metric so that researchers can agree to all use a common metric.
- would be more practically relevant if the experiments were beyond just C_8 rotations on mnist, but closer to SOTA classifiers on imagenet with practically used augmentations such as rotations with arbitrary angles, cropping, flipping, colour distortions, shears etc.
- related point to above - I don't see what is preventing you from going beyond finite groups to generalize the EED metric e.g. to non-group transformations that need not be finite such as those transformations mentioned above. In Eq (3), you could approximate the integral wrt the set of transformations G with samples (i.e. MC approximation) as you are already doing for approximating the integral wrt the data distribution.

[1] SE(3)-Transformers - Fuchs et al., 2020
[2] LieTransformer: Equivariant self-attention for Lie Groups - Hutchinson et al, 2020.
[3] Efficient Generalized Spherical CNNs - Cobb et al., 2021

---

> ### Author Response · Authors · 2022-08-02
> **Reply to Reviewer 9Wpf**
>
> We would like to thank the reviewer for taking the time to closely read our paper and provide helpful feedback. We have tried to address as many of the reviewer’s points as we could below:
>
> “The equivariance error metric in the paper is very much related to those used in various previous works on group equivariance, so not entirely novel. Search for 'equivariance error' in e.g. [1-3].”
>
> Thank you for pointing this out. We have now added citations to these papers in our related works section and amended some of our language in the introduction. We see our paper as adding additional value to the community in a few ways: (i) we develop our $G$-EED metrics from a solid group-theoretic basis and design them to be agnostic to the particular symmetry group or datatype and (ii) we use our metrics to study how deep learning models learn (or do not learn) invariance and equivariance generally. This is in contrast to other works whose metrics are used to evaluate a specific model.
>
> “would be more practically relevant if the experiments were beyond just C_8 rotations on mnist, but closer to SOTA classifiers on imagenet with practically used augmentations such as rotations with arbitrary angles, cropping, flipping, colour distortions, shears etc.”
>
> We agree with the reviewer that more results in line with this comment would have made the work stronger. In this direction we have now added Section A.4, which contains latent space $G$-EED results for various well-trained ResNet50 models (including a BiT model and models trained with self-supervised approaches such as SimCLR and DINO) with respect to reflection across the vertical axis of ImageNet images.
>
> “related point to above - I don't see what is preventing you from going beyond finite groups to generalize the EED metric e.g. to non-group transformations that need not be finite such as those transformations mentioned above. In Eq (3), you could approximate the integral wrt the set of transformations G with samples (i.e. MC approximation) as you are already doing for approximating the integral wrt the data distribution.”
>
> We totally agree with this point. We haven’t gotten to it yet but testing our metric against (non-discrete) Lie groups is one of our goals. We agree with the reviewer that randomly sampling from the uniform distribution (with respect to Haar measure) on such groups would be a reasonable first approach to approximating Eq (3).
>
> “It's confusing that the channelwise G-EED is different from softmax & latent space EED, in that a higher value means the model is more equivariant. Perhaps it's more consistent to use negative cosine similarity to be consistent with other metrics.”
>
> Thank you, this is a useful suggestion. We have changed our definition so that if large m(x_1,x_2) indicate that x_1 and x_2 are closer, then we use -m(x_1,x_2) so that bigger $G$-EED always means less invariant/equivariant.
>
> “Why is L2 distance used for latent space EED whereas cosine similarity is used for channelwise EED? If cosine similarity is also used for the former, it might not be necessary to normalizie by M, the average L2 distance between points.”
>
> This is a great suggestion. We have now added additional experimental results for a version of latent space $G$-EED that uses cosine similarity. This yielded some surprising results. For example, we found that the $C_8$-equivariant CNN models that we experimented with were less invariant than standard CNNs when latent space $G$-EED was measured using cosine similarity. Upon reflection we conjectured two reasons for this. The first is based on the observation that all the equivariant architectures that we are aware of are designed to be equivariant on the nose, not up to scaling. This may cause misalignment between the hardcoded constraints of the network (which are not up to scaling) and the metric we are using (which is). In cases where cosine similarity is important, we believe one way to align equivariant architectures might be to utilize projective representations (a type of group representation where equality holds up to scaling). Our second possible reason for this observation arises because of Figure 7, which shows that while $C_8$-equivariant CNNs cluster orbits of points closer together in their latent space (in terms of cosine similarity), conventional CNNs do a better job of spreading distinct orbits away from each other (this is why conventional CNNs show lower latent space $C_8$-EED after normalization).
>
> “Figure 2 is used as evidence to claim "learned invariance does not arise through learned equivariance". It would be more convincing to show the plot for each value of k between 1 and 6 as opposed to just 2 & 5. As you mentioned in Prop 3.1, it could be that the first 5 layers aren't equivariant but all 6 conv layers are equivariant.”
>
> This is a good point. We made that remark based on review of these other layers. We have now made these plots available to the reader in the Appendix (Figure 6).

---

> > ### Author Response · Authors · 2022-08-02
> > **Reply to Reviewer 9Wpf (cont.)**
> >
> > “In line 277-278, you mention that "invariance is useful when it is aligned with other metrics more directly related to the task the model was designed for". This makes me curious as to what would happen if you use each EED metric as a regulariser for a CNN - would it help of harm generalization?”
> >
> > This is a fantastic idea, and something we now plan to try in the future.
> >
> > “(minor) Figure 4 - x axis shifted by one.”
> >
> > Thank you, we plan to work to make the labels more intuitively line up with the bars in this chart as we revise.

---

> > > ### Comment · Reviewer_9Wpf · 2022-08-07
> > > **Response to rebuttal**
> > >
> > > I appreciate the authors' hard work in their response to the review. I will raise my score accordingly.

---

### Official Review · Reviewer_1AHW · 2022-07-07

**Rating:** 7
**Confidence:** 4
**Soundness:** 3 good
**Presentation:** 3 good
**Contribution:** 3 good

**Summary:**

This paper proposes a method for empirically measuring the degree of equivariance of a CNN to a given dataset.  In experiments on real datasets, the paper finds that equivariant architectures (but not data augmentation) does lead to equivariant layers, that equivariant architectures do not always lead to invariant final layers (perhaps due to accumulation of artifacts), and that use of pretrained weights may prevent the learning of equivariant layers.

**Questions:**

1) How can one ensure that the group action is defined in a way that could capture any hidden group structure in a non-equivariant architecture?  Wouldn't it be necessary to search over possible group structures within a layer?
2) How did you define the C8-group actions for the standard CNN and for the steerable CNN?
3) How would you define the appropriate group actions for a more complicated example, such as 3d rotations for volumetric data?
4) Line 538: How does it follow from the assumptions of Proposition 3.2 that the action of G on Y is trivial?
5) Appendix A.4: How do the (if) and (only if) directions in Proposition 3.2 follow from measurability of h, non-negativity of h, m, and from (6)?  Could you write out these steps of the proof explicitly?


**Limitations:**

The paper addressed a number of important limitations, but does not address the serious problem of how to correctly define the group actions, especially when there could exist hidden approximate group structure that was not specified by the architecture but rather emerged during training.

**Strengths And Weaknesses:**

## Originality
While many methods have been developed for CNNs to be built in a manner to be exactly or approximately equivariant, it is a challenging problem to assess the degree of equivariance in neural net architectures that are not a priori equivariant.  To my knowledge, this paper is the first to attempt to do so.
## Quality
The question of how to define an approximately equivariant layer, even in a network not explicitly designed to be equivariant to a particular group, is a very interesting question.  It has implications for the entire subfield of equivariant neural networks, and so I think it is important to investigate this question thoroughly.  Therefore, before I present any criticisms I might have about the paper, I would like to personally encourage the authors to not be discouraged but rather to deepen their investigation, as it could have impact on how much the community invests into researching equivariant CNNs in the future.

Given the claim of the paper that "data augmentation does not lead to equivariant layers", it is crucial that we have a clear definition of "equivariant layer."  The notion used by the paper, roughly stated, is that an (approximately) equivariant layer is one where for any transformation of the the input taken from the group G, there exists a corresponding group action on the layer which renders the result to be approximately equivalent to the output of the layer on the untransformed input.  However, within the definition there is some ambiguity to what latitude one has to define the group action.  Different approaches to defining the group action may either reveal or fail to reveal an equivariant structure.  Therefore, in order for this paper to succeed at answering the question "how to measure degree of equivariance," I think it is crucial to tighten the definition of approximate equivariance developed by this paper.  In particular, it needs to be decided:
* Is an equivariant network one where there exists *some* group action that achieves close to the optimal value of Equation 3?
* Or is equivariance a property of a *pair* (network, group action), so that we cannot say that "this network is equivariant" or "this network is not equivariant", but only that "this network is (not) equivariant *with respect to* this particular group action"?

My guess (given lack of details) is that for the standard CNN, the group action employed was one that produces a rotation of the output of the layer given a rotation of the input.  However, this is not the only group action that might make sense for the CNN!  You see, I am concerned that the proposed approach is failing to detect more complex equivariant structures that might emerge during training.  I do not know these hypothesized equivariant structures actually occur in practice.  However, I think it is a crucial question to check whether they exist.  I will now proceed to describe the possible hidden equivariant structures that I think the proposed approach might be missing.  In short, I think a structure similar to what equivariant CNNs are designed to learn might emerge naturally during the training of a non-equivariant CNN.

I call these hypothetical structures *emergent equivariant stacks*.  In fact, Cohen and Welling (2016) first broach the possibility of such an equivariant structure, as they state on page 5 "if an ordinary CNN learns rotated copies of the same filter, the stack of feature maps is equivariant, although individual feature maps are not."  To illustrate my understanding of what an equivariant stack of feature maps might look like, consider the following.  Let $\{e, r, r^2, r^3\}$ be the elements of the C4 group (90-degree rotations).  An equivariant stack would consist of four feature maps, labeled $\phi_0, \phi_1, \phi_2, \phi_3$, where $\phi_i(X) \approx r \phi(r^{-i} X)$.  Without loss of generality, suppose the layer $f$ consists of these four neurons, and the output of $f$ is

$$f(X) = (\phi_0(X), \phi_1(X), \phi_2(X), \phi_3(X))$$

Then for the proposed approach to capture the approximate equivariance of the network, the group action would have to be defined such that

$$r f(X) = (\phi_1(r X), \phi_2(r^{-1} X), \phi_3(r^{-1} X), \phi_0(r^{-1} X))$$
$$r^2 f(X) = (\phi_2(r^{-2} X), \phi_3(r^{-2} X), \phi_0(r^{-2} X), \phi_1(r^{-2} X))$$
$$r^3 f(X) = (\phi_3(r^{-3} X), \phi_0(r^{-3} X), \phi_1(r^{-3} X), \phi_2(r^{-3} X)).$$

Since the group structure of $\{\phi_0, \phi_1, \phi_2, \phi_3\}$ would be unknown a priori, to identify such a hidden equivariant structure, one approach might be to search over permutations of the neurons as part of the process of defining the group action on the layer.  Therefore, perhaps the proposed approach could be augmented by including such a group structure, leading to a stronger notion of equivariance (rather than equivariance with respect to a particular pre-defined group action), and which I would consider a far more comprehensive way to answer the question "does data augmentation lead to equivariant layers?"  Note also that the framework of the paper (equations 2 and 3) would be more or less kept intact: the search for group structure would be a valuable add-on to the existing material in the paper.

I personally think that it is essential for the paper to look into the possibility of emergent equivariant stacks.  That is why I have recommended borderline rejection for the current submission.  However, it is possible that reviewers with more experience with equivariant networks may disagree with my assessment of the plausibility or the importance of accounting for emergent equivariant stacks.  In that case, I could easily see them recommending a higher score.

Now, it could also very well be the case that equivariant stacks of more than one feature are *not* likely to emerge for non-equivariant CNNs.  I think it is important to check.  But if the problem I outlined does not actually matter in practice, then the approach adopted by the paper is fine.  Conditional on hidden equivariance not being an issue, there are many positive aspects of the paper.  The measure of equivariance (eqs 2 and 3) appears to be sound, and the evaluation is reasonable and produces interesting results.  The quality would be higher if more types of symmetry groups and data types were evaluated, such as a simple p4 G-CNN, where the method should show exact equivariance, and a more complicated example such as 3d rotations for point clouds.
## Clarity
The paper is clearly written, but I found one of the proofs to be lacking in details.  It would also greatly improve the clarity of the paper if the example presented (C8) could be explained in detail.  The difficulty in understanding exactly how the method operates in a non-elementary concrete example, such as C8, lowers the clarity of the paper.
## Significance
As I stated in "Quality", I think the question being investigated is very important, as it could have impact on how much the community invests into researching equivariant CNNs in the future.  While it is unclear how well the method can be applied to non-equivariant CNN architectures, the method seems to be reasonable for equivariant architectures, and is immediately useful for evaluating the success of equivariant architectures. The finding that C8 has lower equivariance for later layers is potentially novel and interesting, although details to reproduce the finding are currently missing.
## Suggestions
I would ideally like to see an approach similar to the paper that could include a strategy to look for emergent equivariant stacks.  However, I acknowledge that the authors are not likely to have enough time to implement this during the response period.  I would be willing to increase my score to 'borderline accept' or 'weak accept' if the authors can discuss the limitation of the current method in being insensitive to stacks of approximately equivariant features, and also include missing details that would increase the reproducibility of the experiments.  In that case, publication of the paper could serve to stimulate further research in measuring equivariance in networks.

---

> ### Author Response · Authors · 2022-08-02
> **Reply to Reviewer 1AHW**
>
> We would like to thank the reviewer for a very insightful review. The weakness that is one of the central points of this review (that we implicitly assume a particular group action on hidden representations) is quite important and is something that we lost track of in the process of writing this work. We feel that bringing this point to the fore is an important step in improving the paper.
>
> “Is an equivariant network one where there exists some group action that achieves close to the optimal value of Equation 3? Or is equivariance a property of a pair (network, group action), so that we cannot say that "this network is equivariant" or "this network is not equivariant", but only that "this network is (not) equivariant with respect to this particular group action"?
>
> Great question. We implicitly meant the latter. We have rewritten some sections to make this distinction clearer.
>
> “The quality would be higher if more types of symmetry groups and data types were evaluated, such as a simple p4 G-CNN, where the method should show exact equivariance, and a more complicated example such as 3d rotations for point clouds.”
>
> This is a good point. We have now included two additional groups, an elementary reflection group (reflection across the vertical axis of an image) and the dihedral group $D_8$ generated by a rotation of 45 degrees and a reflection across an axis. We agree that including datatypes beyond images (such as point clouds or graphs) would make a future version of the paper stronger.
>
> “I personally think that it is essential for the paper to look into the possibility of emergent equivariant stacks.”
>
> My co-authors and I really loved this idea and believe that it could serve as the basis for a much more interesting and important line of research in the future. The computational algebra community has techniques for illuminating unknown group actions that it might be possible to leverage, unfortunately we were not able to do any of this in the time we had.
>
> As a tiny step forward, we observed that many of the human-designed equivariant CNN architectures (which use the regular representation) utilize group actions on hidden representations that not only act on individual channels (e.g., rotate each channel) but also permute the channels. That is, a group not only acts on individual channels in a 3-tensor, it also permutes these channels in some highly structured way. Therefore, when searching for unknown group actions on a hidden space, the first step might be to remove the assumption that when comparing $f(gx)$ and $gf(x)$, one should compare channel 1 of $f(gx)$, with channel 1 of gf(x), channel 2 of f(gx) with channel 2 of gf(x), etc.
>
> Rather than testing many different individual input and output, in Section A.5, we chose to look at a subset of individual filter channels $w$ in a sample AlexNet model. For each of these $w$ we searched the network for learned elements of $Gw$ (the orbit of w under a specified action of $G$). Our initial search suggested that at least for those AlexNet models that we studied, this type of learned equivariance was not present. This is admittedly a very small step since it is still a very restrictive form of equivariance (though not as restrictive as the definition of channelwise $G$-EED). In particular, we are still assuming a specific action of the group on hidden space (a combination of predetermined actions on individual channels plus some unknown permutation of those channels). Even in the case of human-designed equivariant networks, this type of structure only arises when the regular representation is used. We hope to push this direction of study further in future work.
>
> “I would ideally like to see an approach similar to the paper that could include a strategy to look for emergent equivariant stacks.”
>
> This is an excellent idea. We now have a very small first step towards generalizing the types of group action that can be detected in a network’s hidden representation in Section A.5. Of course, most the work in this direction has yet to be done.
>
> “How did you define the C8-group actions for the standard CNN and for the steerable CNN?”
>
> In this work, the action is always assumed to be the same as the action on the input channels. For example, when the group is $C_8$, the action on hidden representations is rotation of each channel of the hidden representation by multiples of 45-degree angles. We have added language to Section 3.1 to clarify this. We note that when we compare output channels for $C_8$-equivariant models, we compare channels based off our knowledge of how elements of $C_8$ permute the channels. This discrepancy was part of the impetus (along with reviewer feedback) for the new Section A.5.

---

> > ### Author Response · Authors · 2022-08-02
> > **Reply to Reviewer 1AHW (cont.)**
> >
> > “How would you define the appropriate group actions for a more complicated example, such as 3d rotations for volumetric data?”
> >
> > This is a great question, and one that we don’t cover directly in our paper. As the reviewer probably noticed, in all the examples that we cover the output datatype is either equivalent up to changes in dimension (e.g., for images the input is a 3-tensor and the output is a 3-tensor with possibly a different number of channels and spatial dimensions) or is a very common output type like a vector (from a softmax layer or a latent space). It would be interesting to consider more complicated cases that do not fall into these two categories, but we have left this to future work where we look at examples of equivariance in ML beyond image recognition.
> >
> > “Line 538: How does it follow from the assumptions of Proposition 3.2 that the action of G on Y is trivial?”
> >
> > Thank you for pointing this out, we had forgotten to update the statement of the Proposition and the proof from a previous version. These have now been corrected. Note that we have included an extra assumption that $G$ acts faithfully on $X$ and $Y$.
> >
> > “Appendix A.4: How do the (if) and (only if) directions in Proposition 3.2 follow from measurability of h, non-negativity of h, m, and from (6)? Could you write out these steps of the proof explicitly?”
> >
> > Thanks for the feedback. In the process of rewriting the proof for Prop. 3.2 we also tried to make the two-way implication clearer.

---

> > > ### Comment · Reviewer_1AHW · 2022-08-03
> > > **Nice work**
> > >
> > > Dear authors,
> > >
> > > I am impressed by the results you were able to put together in a short time to look into the "emergent equivariant stacks" and I look forward to further developments.  It also looks like you were able to address the issues in the proofs.  Finally, the additional detail about implementation and additional examples of symmetries improves the practical applicability of this paper.  I am happy to raise my score to 7.

---

### Official Review · Reviewer_YrEP · 2022-07-07

**Rating:** 6
**Confidence:** 3
**Soundness:** 2 fair
**Presentation:** 3 good
**Contribution:** 1 poor

**Summary:**

The paper aims to formally define group-theoretic family of metrics to evaluate empirical equivariance of a model. The contributions can be summarized as
- the introduction of a G-empirical equivariance deviation (G-EED)
- the introduction of possible instantiations of such deviation to measure specific invariances and equivariances
- the use of G-EED to obtain insights on equivariance and invariance in neural networks


**Questions:**

- G-EED goodness strictly depends on the choice of the finite group.
For example, if we were to test against any possible rotation, the cyclic C_8 group the authors selected might not be sufficient in fully evaluating invariance to rotation.
Similarly in the context of permutation invariance (which would be measured at the latent space and softmax level) one would ideally try all the possible permutations over the input set.
However, to experimentally evaluate G-EED we need to provide a finite group. Could the authors comment on how to select the most suitable group in practice?  How does the choice of the group impact the reliability of the deviation? A discussion about regular spacing vs random sampling would be interesting as well in selecting the finite rotation group. What is the best spacing? And/or what is the best number of sampled rotations? Are uniform samples always the best options or are other possible sampling strategies?
A possible experiment could be to show how G-EED changes by adding more rotation angles to C_8.
- The authors write in line 277-278 “Invariance is useful when it is aligned with other metrics more directly related to the task the model was designed for”, yet in their motivation they state that understanding invariance allows them to detect model failures. While it seems that G-EED allows for interesting comparison over different models designed for similar tasks, wouldn't the authors agree that to actually evaluate failure and success G-EED provides less of a tool than standard metrics?

- It could be good to have an intuition of the deviation range to be able to use G-EED on a new model. Ideally, one would like to know that  below (or above) a certain value the G-equivariance of the network is good. Do the authors have a way to estimate such value? If not, would the authors say that the proposed deviation has to always be used comparatively?
To make a more practical example, the right panel of Figure 3 shows a high variability in the softmax for CNN. The plot intuitively seems to suggest that CNN has  worse performance, yet the classification results are good >98%. To my understanding, this means that the relative distance of the two models compared to the overall possible range of the deviation is not that significant.
- Figure 2 shows that as iterations go by the equivariance properties of CNN decreases, the authors propose as an explanation the fact that inductive bias on equivariance might not be aligned with learning objective, yet, in a later experiment they show that random weights pertain higher equivariance than transfer weights, could the decay then simply due to initialization?
- The discrepancies between latent space and softmax suggest that the decoder has a high importance in learning invariances. If the network is $f = f2 \circ f1$, the results seem to suggest that even if $f1$ is not G-equivariant $f$ might be through the mean of the decoder $f2$. However, generally, in the construction of architectures more importance is given in encoding equivariance in early layers. If I'm interpreting the results correctly, this could actually mean that such efforts are misplaced. Do the authors agree? Could they comment on this?

Please improve the figures, for example by making the text bigger and improving definition as well as aligning the ticks in Figure 4 to improve readability.

Typos:

- Line 134 (then instead of than)

- line 162 where


**Limitations:**

The authors address some limitations in the paper. I would have liked a broader discussion on the usability of the proposed approach given the many prior choices that G-EED depends on. I'm happy to increase my score based on authors’ response and if other more expert reviewers have better feedback.


**Strengths And Weaknesses:**

The paper is well written and easy to follow. I found particularly pleasant the formal introduction of G-EED and its possible instantiations to measure equivariance and invariance at different parts of the network. The intuitions provided by the authors about different equivariant and invariant mechanisms are interesting. However,  the experimental validation of the proposed approach is somewhat lacking and the paper could have higher impact with a more well-thought experiment and result section. The authors test their deviation just in the context of 45 degrees rotation. It would make a more compelling argument of the usefulness of the proposed deviation to analyze other invariances, e.g. translation, permutation.

---

> ### Author Response · Authors · 2022-08-02
> **Reply to Reviewer YrEP**
>
> We would like to begin by thanking the reviewer for their thoughtful comments. It is clear that the reviewer considered both the paper and the implication of its results carefully which we appreciate. We have attempted to respond to the reviewer’s comments and questions below:
>
> “G-EED goodness strictly depends on the choice of the finite group…”
>
> Our understanding of this question is that the reviewer is asking about approximation of the invariance with respect to a group $G$ by either computing $H$-EED where $H$ is a subgroup of $G$ or by randomly sampling from $G$. This is a valuable direction of research and one that we hope to explore in future work. One would certainly have to do some type of sampling if one wanted to compute $G$-EED where $G$ is a (non-discrete) Lie group. It would be interesting to understand if there are effectively any differences between the case where one  samples uniformly at random from $G$ with respect to the Haar measure rather than choosing a finite subgroup contained in $G$ and sampling from that. This would illuminate whether preserving algebraic structure is important to approximating $G$-EED.
>
> “The authors write in line 277-278 “Invariance is useful when it is aligned with other metrics more directly related to the task the model was designed for”, yet in their motivation they state that understanding invariance allows them to detect model failures. While it seems that G-EED allows for interesting comparison over different models designed for similar tasks, wouldn't the authors agree that to actually evaluate failure and success G-EED provides less of a tool than standard metrics?”
>
> We would certainly admit that the most important metric is the metric that directly addresses the goal of the task and that this is the first place that one should start when evaluating a model. On the other hand, in real-world tasks, there can often be multiple metrics that are important to a model developer. In safety-critical applications for example, it is important to understand whether a model will fail because of non-semantically meaningful changes to input (such as changes in orientation, reflection, etc.) Confirming that this is not likely to happen for specific types of transformations (using a metric like $G$-EED) does not assure us that there are not other types of non-meaningful transformations that a model fails on, but we at least understand the landscape of model failure a little better.
>
> “It could be good to have an intuition of the deviation range to be able to use G-EED on a new model. Ideally, one would like to know that below (or above) a certain value the G-equivariance of the network is good. Do the authors have a way to estimate such value? If not, would the authors say that the proposed deviation has to always be used comparatively? To make a more practical example, the right panel of Figure 3 shows a high variability in the softmax for CNN. The plot intuitively seems to suggest that CNN has worse performance, yet the classification results are good >98%. To my understanding, this means that the relative distance of the two models compared to the overall possible range of the deviation is not that significant.”
>
> This is a great question. We expect that our metrics will generally work best when used in a comparative fashion. Indeed, we have a hard time imagining how one could compute a threshold for a metric such as latent space $G$-EED without a comprehensive understanding of the dataset in question. We have included this observation in our limitations section.
>
> “Figure 2 shows that as iterations go by the equivariance properties of CNN decreases, the authors propose as an explanation the fact that inductive bias on equivariance might not be aligned with learning objective, yet, in a later experiment they show that random weights pertain higher equivariance than transfer weights, could the decay then simply due to initialization?”
>
> Thank you for pointing this out. We had not connected this particular observation with our claim about the alignment of the loss function and equivariance. It is possible that a similar phenomenon is occurring here.

---

> > ### Author Response · Authors · 2022-08-02
> > **Reply to Reviewer YrEP (cont.)**
> >
> > “The discrepancies between latent space and softmax suggest that the decoder has a high importance in learning invariances. If the network is f=f2∘f1, the results seem to suggest that even if f1 is not G-equivariant f might be through the mean of the decoder f2. However, generally, in the construction of architectures more importance is given in encoding equivariance in early layers. If I'm interpreting the results correctly, this could actually mean that such efforts are misplaced. Do the authors agree? Could they comment on this?”
> >
> > This is an excellent observation. We were surprised by this as well and believe that it suggests (as the reviewer pointed out), that in non-equivariant networks, the final layer(s) may play an important role in total model invariance. We have a list of additional experiments that we would like to run that would further illuminate which layers are most important to learning invariance in a network. These would involve training a network on a subset of transformations (e.g., only MNIST digits in the usual orientation), then freezing different parts of the network and continuing to train on the broader range of transformed examples. This type of ablation study might help distinguish which parts of the network are most important for overall invariance.
> >
> > “Please improve the figures, for example by making the text bigger and improving definition as well as aligning the ticks in Figure 4 to improve readability.”
> >
> > Thank you for the feedback, we plan to reformat these plots at the next opportunity for additional polishing.
> >
> > “Line 134 (then instead of than)”
> >
> > Thank you, this has now been corrected.
> >
> > “line 162 where”
> >
> > We actually meant to use “were” in this case. We have rephrased to make the wording clearer.

---

> > > ### Comment · Reviewer_YrEP · 2022-08-08
> > > **Thanks!**
> > >
> > > I'm happy with the authors response and I updated my score.

---

### Official Review · Reviewer_Lyvv · 2022-07-11

**Rating:** 5
**Confidence:** 4
**Soundness:** 2 fair
**Presentation:** 3 good
**Contribution:** 2 fair

**Summary:**

This paper proposes a measure of equivariance (and invariance) in 3 different contexts for multi-class classifiers: (1) at channel level in a CNN, (2) at representation level and (2) at softmax level.

The proposed measure, generically called *G*-EED, is derived from the equivariance requirements of a mapping $f$ made up of the compositions of several mappings $f_1, \ldots, f_n$. The derivation is clear and insightful, I appreciated the that the mathematical derivation is well written and can be followed nicely.

More precisely, this work uses the generic formulation of *G*-EED to assess how **equivariant** intermediate layers of a CNN are, and how **invariant** the representation and softmax outputs are. The reason being that input transformation acts faithfully on intermediate spatial feature maps (thus allowing to measure equivariance), but are harder to transfer to representation and softmax (thus choosing to measure invariance).

One interesting aspect is that results are obtained using few test samples (50), which adds value to the method.

The results focus on using *G*-EED to assess how equivariance (and invariance) evolves during training, comparing a regular CNN and a group equivariant CNN for the C8 group (C8-CNN). Several OOD datasets are tested, yielding interesting findings related to how equivariance is preserved across layers and during training, as well as invariance properties of different DNNs and different initialization strategies.

Overall, the paper is well written, with clear language and is enjoyable to read.

**Questions:**

* What is the justification for the use of xView Maritime? Is it because it has features in all orientations? If so, I'd specifically point this out in the paper.

* The $Ker(\phi_Y)$ definition in L153-154 is interesting. However,  I wonder if it is strictly necessary, since later on such  $Ker(\phi_Y)$ is assumed to be the full $G$: all groups considered in the paper either act faithfully (for channelwise) or the $g$ elements are not required (for latent and softmax).
  * I believe the paper would benefit from showcasing a usecase where the use of $Ker(\phi_Y)$ is required. It would also reinforce the derivation of *G*-EED using the $Ker(\phi_Y)$ definition.

Some questions/suggestions related to the analysis of pre-trained vs. scratch models:
* By eye-balling the results, it seems that Resnet converges to almost the same level of invariance with both initialization strategies. Could this be related to the presence of skip connections in that architecture? A similar (milder) result is observed for DenseNet, which also has several skip connections. I'd suggest doing a "channel"-wise analysis of the models shown in Figure 5, which could lead to some better understanding.
* Resnet pre-trained achieves better invariance than LeViT trained from scratch. Is the architecture playing a big role in these results, even more than the pre-training strategy itself?
* I would suggest adding a pre-trained model using some modern Self-Supervised Learning method (eg. SimCLR [1], BYOL [2]) and comparing those features with the ones pre-trained supervised.

Related to the different metrics used in the different *G*-EED variants:
  * What is the justification for the the use of cosine similarity, $\ell_2$ and $D_{KL}$ respectively. See the previous comment in the strengths/weaknesses section.
  * Minor comment: There is a definition of  $\mathcal{E_{latent}}$ and $\mathcal{E_{softmax}}$, for completeness I suggest to add $\mathcal{E_{channel}}$ in the equation after L188.

*References:*
* [1] Chen, Ting, et al. "A simple framework for contrastive learning of visual representations." International conference on machine learning. PMLR, 2020.
* [2] Grill, Jean-Bastien, et al. "Bootstrap your own latent-a new approach to self-supervised learning." Advances in neural information processing systems 33 (2020): 21271-21284.


**Limitations:**


The authors mention the limitations in Section 6, pointing out that the method is applied to classification but could be extended to other ML tasks.
The fact that exact symmetry groups are required is also mentioned as a limitation of the method.


I do not see other societal issues related to this work.

**Strengths And Weaknesses:**

**Strengths**:

* The theoretical aspects of the paper are solid and the derivation of *G*-EED is reasonable.

* The results show interesting findings: (1) Equivariance diminishes as training progresses for all CNNs, (2) Equivariance diminishes as we go deeper in the CNN, (3) C8-CNN yields higher invariance for all OOD datasets tested, (4) random networks can be perfectly invariant, raising a valid point about the assessment of invariance only and (5) pre-trained models show different invariance a properties than models trained from scratch.

* I would like to emphasize the easy implementation of the proposed measure, as well as the low computational requirements (at least with the datasets shown in the paper), only requiring 50 data points.

* The paper is well written, easy to follow and seems a priori reproducible from the data given.

**Weaknesses**:

Listing hereafter some weaknesses I found while reading the paper, and also some suggestions aiming at making the paper even stronger.


* The proposed approach is limited to exact symmetry groups such as rotation or reflection, which act faithfully on the spatial feature maps. While these groups are of great interest to the community, I still believe this is a limitation of the method. The community is actively researching how to tackle equivariance properties for non exact groups such as color augmentations, and any step in that direction would be highly appreciated.

* Although theoretically sound, the metrics proposed eventually get simplified into basic equivariance/invariance losses.

* Only one group (C8) is analyzed. I believe the paper would improve if the analysis was performed for some other exact group.

* I found the use of 3 different metrics $m$ (one per each version of *G*-EED proposed) not as elegant as the rest of the paper. Using the same metric would be preferred. Here some comments about the metrics used in the paper:
  * To compare (flattened) feature maps it seems more appropriate to use $\ell_2$, since the magnitude is important in intermediate layers (their output will be consumed by later layers). By using cosine similarity, magnitude is not considered.
  * Similarly, to compare representations ($Z$), it seems more appropriate to use cosine distance. In this case, the high dimensional representation is not expected to be clustered in terms of absolute distance but in terms of angle. In such high dimensionality, being close-by in terms of angle is already sufficient, since any pair of randomly sampled vectors is (mostly) orthogonal. Therefore, cosine distance is a common choice.
  * The choice of $D_{KL}$ for softmax can also be misleading. Let $f(x)$ be classifier that performs poorly (eg. yields $f(x)=constant+\epsilon$, with $\epsilon \sim \mathcal{N}(0, \sigma)$ and $\sigma$ very small). Then $\mathcal{E_{softmax}}$ will be very low, indicating invariance, since the softmax distribution of $f(x)$ and $f(gx)$ are the same.  However, the classifier is actually poorly invariant ($f(x)$ changes randomly). Summarising, the choice of $D_{KL}$ implicitly assumes a well-trained classifier. This could also partly explain why *G*-EED yields 0 (fully invariant) for the randomly initialised model.
  * I believe the choice of metric is critical in this method and should be evaluated in an ablations study (and theoretically if possible).

* The comparison of pre-trained and scratch models is not as conclusive as the other results in the paper. I believe some additional analysis could be done in that sense.

* Minor comment: I found the schema in Figure 1 hard to interpret. I'd add some extra information to it that more accurately shows what is done at each step in the DNN chain.

---

> ### Author Response · Authors · 2022-08-02
> **Reply to Reviewer Lyvv**
>
> We would like to start by thanking the reviewer for their thoughtful review. We appreciate the feedback and felt that all suggested changes will make the paper stronger if included. We have tried to address as much as possible over the course of the last week. We cover each of the reviewer’s points that we were able to address below. We regret that we did not have time to address every piece of feedback during the rebuttal period.
>
> “The proposed approach is limited to exact symmetry groups such as rotation or reflection, which act faithfully on the spatial feature maps.”
>
> This is an excellent point. We see the main challenge as arising from the fact that we don’t understand exactly what the “correct” action of the “change of hue group” is on a latent space. Perhaps this is what the reviewer meant. Specifically, if $g$ is a shift in hue and $f_1: X \rightarrow X_k$ is the first $k$ layers of the network, we know the action of $g$ on $X$, but we don’t know the action of $g$ on $X_k$.
>
> Our (philosophical) perspective was that it made sense to initially design a metric for linear representations of groups (where extensive, rich theory exists) and then explore how this metric could be generalized to the action of algebraic objects that are not groups (e.g., scaling). We have added an extra line in the limitations section to make this point explicit.
>
> “Although theoretically sound, the metrics proposed eventually get simplified into basic equivariance/invariance losses.”
>
> We felt that the fact that our theory-based constructions actually resulted in metrics that were ‘basic’ was a success. We would have been concerned if our theory-based constructions led us to metrics that were unintuitive.
>
> “Only one group (C8) is analyzed. I believe the paper would improve if the analysis was performed for some other exact group.”
>
> We agree with the reviewer that the paper could be made significantly more compelling if other groups were included. To this end we ran experiments where we calculated latent space $D_8$-EED for the models in Section 4.3, where $D_8$ is the order 16 dihedral group (that is, the group generated by a rotation of 45 degrees and a reflection across one of the coordinate axes). We found similar patterns to those we found for the same models with $G = C_8$.
>
> In Section A.4, we also compared supervised and self-supervised ResNet50s that had been trained on ImageNet in terms of latent space $G$-EED for $G$ equal to reflections across the vertical axis of an image.
>
> Ideally, we would have also investigated how our methods could be applied to (non-discrete) Lie groups where some level of sampling is required. Unfortunately, we have not done this yet. For non-trivial Lie groups, such as $SO(n)$ for $n > 2$, it seems reasonable to sample randomly from the uniform distribution on the manifold (with respect to Haar measure). Algorithms for doing this are well known [1], but require some care when implementing. Our plan is to pursue this direction in future work.
>
> “I found the use of 3 different metrics m (one per each version of G-EED proposed) not as elegant as the rest of the paper. Using the same metric would be preferred.”
>
> Following the same inclination as the reviewer, we initially put significant effort into identifying a single metric that could be used universally across different possible applications of $G$-EED. We agree that this would in many ways be more compelling than asking the user to specify their own metric. However, in our experiments we were unable to find a single metric or distance function that we could foresee working across all the applications we had envisioned. The perspective that we now take is that by not specifying the metric we make $G$-EED more flexible for different model types, modalities, and tasks. This is especially important since in future work we hope that $G$-EED can be calculated for models that are not classifiers (for example object detectors and segmentation models).
>
> “To compare (flattened) feature maps it seems more appropriate to use $\ell_2$, since the magnitude is important in intermediate layers (their output will be consumed by later layers). By using cosine similarity, magnitude is not considered.”
>
> We chose cosine similarity after experimenting with several different metrics along with some representative test cases which we felt did or did not represent equivariance relevant to CNNs. For example, we found that $\ell_2$ distance led to non-equivariant layers being measured as more equivariant than layers with hardcoded equivariance. In these cases, we visually inspected both the input and outputs of both networks and concluded that an effective metric would not have provided this outcome. We agree that in future work it would be worthwhile to include some of these experiments for the reader to explore.

---

> > ### Author Response · Authors · 2022-08-02
> > **Reply to Reviewer Lyvv (cont.)**
> >
> > “Similarly, to compare representations (Z), it seems more appropriate to use cosine distance.”
> >
> > We thank the reviewer for this suggestion since it led to some interesting analysis in Section 4.1 where we now have results for latent space $G$-EED for both equivariant and non-equivariant models. Somewhat surprisingly, we found that the equivariant models that we tested were not more invariant with respect $G$-EED measured using cosine similarity. Upon reflection we suspect that this results from the fact that the equivariant architectures that we are aware of are designed to be equivariant “on the nose”, not up to scaling. This is an interesting observation and suggests some future avenues to be explored in equivariant architectures. Namely, equivariance within the framework of projective representations, which are representations were equality holds precisely up to scaling.
> >
> > We have also included results for latent space $G$-EED is some of our other experiments (e.g., Sections A.3, A.4).
> >
> > “The choice of DKL for softmax can also be misleading.”
> >
> > We appreciate the reviewer’s example of the noisy constant function, f(x) = constant + noise. We agree that this would not be a particularly useful model. However, the standard definition of invariance (that we work from in this paper) says that this function is indeed close to being invariant. Part of the message of our paper is that the ML community may either want to adjust its definition of invariance so as to be more in-line with the tasks a model is meant to solve or accept pathological examples such as the one that the reviewer provided.
> >
> > “Minor comment: I found the schema in Figure 1 hard to interpret. I'd add some extra information to it that more accurately shows what is done at each step in the DNN chain.”
> >
> > Thanks for the feedback, we have modified the diagram to try and make it more intuitive. In particular, we added additional captions and arrows that help the reader understand what is being compared exactly for each type of invariance/equivariance measurement. We also increased the size of the font.
> >
> > “The comparison of pre-trained and scratch models is not as conclusive as the other results in the paper. I believe some additional analysis could be done in that sense.”
> >
> > Upon review of this section, we feel that the reviewer is correct. We have chosen to soften our conclusion substantially indicating that there seems to be important differences between pretrained models and models trained from scratch, but more investigation is required to identify what these are and how they arise.
> >
> > “What is the justification for the use of xView Maritime? Is it because it has features in all orientations? If so, I'd specifically point this out in the paper.”
> >
> > This is a good point. We have added a sentence to Section 4, giving further explanation for why we chose to use xView Maritime.
> >
> > “The Ker(ϕY) definition in L153-154 is interesting. However, I wonder if it is strictly necessary, since later on such Ker(ϕY) is assumed to be the full G: all groups considered in the paper either act faithfully (for channelwise) or the g elements are not required (for latent and softmax).”
> >
> > We felt it was important to have a comprehensive algebraic framework for measuring equivariance and therefore opted to cover the case where the action of $G$ on the input space has a different kernel than the action of $G$ on the output space. We agree with the reviewer that it would be useful to provide a concrete example of this from the perspective of machine learning and aim to do this in future work.
> >
> > “By eye-balling the results, it seems that Resnet converges to almost the same level of invariance with both initialization strategies…” And “Resnet pre-trained achieves better invariance than LeViT trained from scratch…”
> >
> > These are interesting suggestions! We hope to explore the connections between invariance/equivariance and architectural features in future work.
> >
> > “I would suggest adding a pre-trained model using some modern Self-Supervised Learning method (eg. SimCLR [1], BYOL [2]) and comparing those features with the ones pre-trained supervised.”
> >
> > This was an excellent suggestion. We have now included a section that compares ResNet50 models trained with supervised vs self-supervised training methods. We find that while models trained via contrastive learning methods achieve higher invariance, other self-supervised methods (e.g. DINO) need not be more invariant than models trained via supervised approaches.
> >
> > “Minor comment: There is a definition of Elatent and Esoftmax, for completeness I suggest to add Echannel in the equation after L188.”
> >
> > Thank you, we have added this equation for clarity.

---

> > > ### Comment · Reviewer_Lyvv · 2022-08-04
> > > **A great rebuttal**
> > >
> > > I would like to congratulate the authors for an excellent rebuttal, to my initial comments and to other reviewers. I appreciate the amount of work done in such a short time and how the paper has improved with the new additions.
> > >
> > > I believe that the paper is now much stronger, and of greater interest for the community. I will upgrade my score accordingly.

---

### Author Response · Authors · 2022-08-02
**Overview of paper changes based on reviewer feedback.**

We would like to start by thanking all the reviewers for their helpful and thoughtful reviews. We really appreciate the reviewers taking the time to read our paper closely and provide constructive feedback. We have tried to address all the reviewer comments that we could. We responded to specific feedback in our reply to each reviewer, but we wanted to summarize some of the significant changes that we made to the paper here (in particular that we slightly changed the definition of the metric in line with reviewer feedback):

- Based on a suggestion from reviewer 9Wpf, we now attach a negative sign when using a distance function $m$ where larger values of $m(x_1,x_2)$ indicate that $x_1$ and $x_2$ are “closer”. This way, large values of $G$-EED consistently indicate that a model is less invariant, rather than this varying depending on the $m$ used.
- We ran our metrics against two additional finite groups: the dihedral group of order 16 (rotations and reflections) on xView Maritime (Section 4.3) and the reflection group of order 2 (reflection across the vertical axis) on ImageNet (Section A.4).
- We added a section where we measured the invariance of four ResNet50’s, two trained via supervised methods and two trained with self-supervised methods, with respect to the group of reflections across the vertical axis of ImageNet images (Section A.4).
- We added a section containing preliminary investigation of possible ‘emergent equivariance’ (to use terminology suggested by reviewer 1AHW), where we study whether networks might learn equivariance in convolutional layers both in terms of individual channels and the ordering of those channels (Section A.5).
- We added a short section where we compared invariance for networks trained with and without augmentation (Section A.3).
- We now include experiments for latent space $G$-EED using both Euclidean distance and cosine similarity. This led to some interesting and new conclusions and questions in Section 4.1.

---

### Meta-Review · Area_Chair_QUHy · 2022-08-29

**Recommendation:** Accept
**Confidence:** Certain

**Metareview:**

The paper proposes metrics to empirically explore the nature of invariance and equivariance of deep learning models with the goal of better understanding the ways that they actually capture these concepts on a formal level. They utilize their proposed metrics to shed light on two popular methods used to build invariance into networks, data augmentation and equivariant layers. The reviewers agree on the significance of the contribution and quality of the presentation. Some questions were raised by the reviewers that required some clarification and the the authors did a great job addressing these, leading to  improvements in the paper and as a result in the reviewer’s opinion of the paper.

**Award:**

No

---

### Decision · Program_Chairs · 2022-09-14

Accept